# Additive Models Explained:
# A Computational Complexity Approach

**Shahaf Bassan**
The Hebrew University of Jerusalem
shahaf.bassan@mail.huji.ac.il

**Michal Moshkovitz**
Google Research
mmichal@google.com

**Guy Katz**
The Hebrew University of Jerusalem
g.katz@mail.huji.ac.il

## Abstract

Generalized Additive Models (GAMs) are commonly considered *interpretable* within the ML community, as their structure makes the relationship between inputs and outputs relatively understandable. Therefore, it may seem natural to hypothesize that obtaining meaningful explanations for GAMs could be performed efficiently and would not be computationally infeasible. In this work, we challenge this hypothesis by analyzing the *computational complexity* of generating different explanations for various forms of GAMs across multiple contexts. Our analysis reveals a surprisingly diverse landscape of both positive and negative complexity outcomes. Particularly, under standard complexity assumptions such as P≠NP, we establish several key findings: (i) in stark contrast to many other common ML models, the complexity of generating explanations for GAMs is heavily influenced by the structure of the input space; (ii) the complexity of explaining GAMs varies significantly with the types of component models used — but interestingly, these differences only emerge under specific input domain settings; (iii) significant complexity distinctions appear for obtaining explanations in regression tasks versus classification tasks in GAMs; and (iv) expressing complex models like neural networks additively (e.g., as neural additive models) can make them easier to explain, though interestingly, this benefit appears only for certain explanation methods and input domains. Collectively, these results shed light on the feasibility of computing diverse explanations for GAMs, offering a rigorous theoretical picture of the conditions under which such computations are possible or provably hard.

## 1 Introduction

Generalized additive models (GAMs) are a widely utilized family of models in ML [51, 88], valued commonly for their interpretability [85, 37, 113, 25, 38, 74, 29, 48, 40, 84, 87, 47, 114] due to their additive structure. Beyond being considered interpretable, the very assumption of "additive interpretability" is a foundational one in the explainable AI community. For instance, many explanation techniques are designed to approximate additive behaviors [94], and metrics such as infidelity [112] assess explanation quality by measuring how well an additive approximation captures the effects of feature perturbations on the model's output. Furthermore, interpretable training approaches often seek to promote near-additive behavior in models within specific domains [4].

Since GAMs are commonly regarded as an interpretable class of ML models, it may seem natural to expect that different types of explanations for their predictions can be computed efficiently, without encountering drastic computational intractability (e.g., NP-hardness, #P-hardness, etc.). This

39th Conference on Neural Information Processing Systems (NeurIPS 2025).

question also relates to a growing body of research in the ML community focused on understanding the computational complexity involved in generating different types of explanations for various model classes [16, 104, 18]. These efforts span both inherently complex models like neural networks [1] and tree ensembles [12], where explanations are expected to be computationally hard to compute [16, 1], as well as traditionally "interpretable" models such as decision trees [9] and monotonic classifiers [78], where computing explanations is expected to be more efficient [16, 80, 18].

**Our contributions.** In this work, we investigate the computational complexity of generating various types of explanations for different classes of GAMs across a range of contexts. We also question the assumption that such explanations are inherently efficient to compute, given the perceived interpretability of GAMs. Through a comprehensive analysis spanning a wide array of settings, we uncover a surprisingly diverse landscape — revealing both tractable (efficient) and intractable (hard) cases. Our findings reveal that the complexity of generating explanations is heavily influenced by several factors — including some unexpected ones that were not known to impact complexity in previously studied ML models.

Our analysis is structured around the following three dimensions:

1. **Component model type used within the GAM:** Here, we investigate several popular component models that are typically used in GAMs and span the interpretability spectrum, including: (i) *Smooth GAMs* — i.e., the classical use of splines [51]; (ii) *Neural Additive Models (NAMs)* [3] — where each component is a neural network; and (iii) *Explainable Boosting Machines (EBMs)* [37] — which use boosted tree ensembles as their components.

2. **Type of explanation considered:** We study the complexity of generating a diverse set of explanation types, including: (i) *Minimum sufficient reasons* — a common feature selection task that considers selecting the minimal subset of features that satisfy the common *sufficiency* criterion (as well as two additional relaxations of this task); (ii) *Minimum contrastive explanations* — minimal changes to the input that would alter the prediction; (iii) *Shapley value attributions* — assigning importance scores to features; and (iv) *Feature redundancy identification* — detecting redundant feature contributions.

3. **Input domain setting:** We study three general types of input domains: (i) *Enumerable discrete domain* — where each input variable takes values from a constant set; (ii) *General discrete domain* — where input values are discrete but not necessarily enumerable; (iii) *Continuous domain* — where inputs take real-valued features.

We prove complexity results for generating explanations across all configurations of these dimensions — i.e., across every combination of *component type* × *explanation type* × *input domain* (see Table 1 for a summary of the results). This uncovers a diverse range of results — including (i) *efficient polynomial-time algorithms*, (ii) *intractability* results (NP-hard, #P-hard, etc.), and (iii) *pseudo-polynomial time* results — cases where the problem is generally intractable but becomes efficiently solvable when the GAM's weight coefficients are encoded in *unary* rather than binary. Intuitively, *pseudo*-polynomial tractability suggests that although generating explanations for a GAM may be intractable, reducing the precision of its coefficient weights can render the explanation task efficient.

**Key Insights.** Beyond the direct contributions that our results offer of both efficient algorithms and intractability outcomes, they also uncover several surprising insights into the computational nature of generating explanations for GAMs, as highlighted below:

- **The complexity of computing explanations for GAMs depends heavily on the input domain, unlike other ML models where a variation based on the input domain is not observed.** We show that for most explanation types we studied — like sufficient explanations, contrastive explanations, and Shapley values — computing explanations becomes exponentially harder in continuous and discrete settings compared to enumerable discrete ones. An interesting exception is the feature redundancy explanation, which is actually exponentially *easier* in the continuous case. This significant sensitivity to the input domain is surprising since it appears to be *unique to additive models*, as other ML models (e.g., decision trees, tree ensembles, neural networks) do not exhibit such diversity in complexity across input domains.

- **The complexity of obtaining explanations for GAMs largely depends on their component models, but this effect interestingly appears only in certain input domains.** We

show that for all our explanation types, there are exponential differences in the complexity of generating explanations for GAMs, depending on the underlying component models (e.g., GAMs with splines are significantly easier to obtain explanations for than NAMs and EBMs). Interestingly, however, we uncover that this complexity distinction arises only in the continuous and general discrete input domains, but not in the enumerable discrete domain.

- **An intriguing complexity separation shows that computing explanations for GAMs used for classification is strictly harder than for regression — but only for SHAP, not other explanations.** We demonstrate this by proving polynomial-time results for the regression setting, in contrast to the classification setting, which is #P-Hard — highlighting a notable complexity separation.

- **Some non-additive models (e.g., neural networks) are harder to obtain explanations over compared to their additive counterparts (e.g., NAMs), but this depends critically on the explanation type and input domain.** We show that models typically hard to interpret — such as tree ensembles and neural networks, where computing explanations is at least NP-Hard — become tractable (e.g., solvable in polynomial time, or belonging to lower complexity classes) in their additive forms (e.g., NAMs, EBMs). Interestingly, however, this phenomenon does not always occur and is highly sensitive to the type of explanation and the input domain.

Due to space limitations, we provide only a brief outline of the proofs for our claims in Section 6, with the full proofs presented in the appendix.

## 2 Preliminaries

**Complexity Classes.** This paper assumes familiarity with basic complexity classes such as polynomial-time (PTIME), and non-deterministic polynomial-time (NP, coNP). We also discuss the class $\Sigma_2^P$, which includes problems solvable in NP with access to a coNP oracle and $\Delta_2^P$, which includes problems solvable in PTIME with access to an NP oracle. Clearly, NP and coNP are contained in $\Delta_2^P$ and $\Sigma_2^P$, and it is also widely believed that NP, coNP $\subsetneq \Delta_2^P \subsetneq \Sigma_2^P$ [10]. Additionally, we cover #P, the class of functions that count the accepting paths of polynomial-time nondeterministic Turing machines. Although not directly comparable since they focus on counting rather than decision problems, it is widely believed that the #P class is strictly "harder" than $\Sigma_2^P$, which can be denoted as $\Sigma_2^P \subsetneq$ #P [10].

**Setting.** We consider a set of input features indexed by $\{1, \ldots, n\}$, where $n \in \mathbb{N}$, and denote an input instance as $\mathbf{x} := (\mathbf{x}_1, \ldots, \mathbf{x}_n)$. Each feature $i$ has a domain $\mathcal{X}_i$, such that the input space is defined as $\mathbb{F} := \mathcal{X}_1 \times \mathcal{X}_2 \times \ldots \times \mathcal{X}_n$. The predictive model $f$ maps inputs from $\mathbb{F}$ either to a regression output, $f : \mathbb{F} \to \mathbb{R}^d$ with dimension $d \in \mathbb{N}$, or to a classification label, $f : \mathbb{F} \to [c]$, where $c \in \mathbb{N}$ is the number of classes. For clarity in our proofs, we restrict $f$ to have a one-dimensional output — i.e., $f : \mathbb{F} \to \mathbb{R}$ in the regression case and $f : \mathbb{F} \to \{0, 1\}$ in the classification case — though we emphasize that our results extend to multi-dimensional outputs (see Appendix D for details). The majority of explanations we study are *local*, aiming to interpret the model's prediction at a specific instance $\mathbf{x} \in \mathbb{F}$ — that is, to explain why $f(\mathbf{x})$ was predicted. However, we also consider some *global* explanations, which aim to interpret the overall behavior of $f$, independent of any particular input.

**Generalized Additive Models (GAMs).** A GAM $f$ is defined as an additive model consisting of $k$ component models $f_1, f_2, \ldots, f_k$. Formally, in the regression setting, we define:

$$f(\mathbf{x}) := \beta_0 + \beta_1 f_1(\mathbf{x}_1) + \beta_2 f_2(\mathbf{x}_2) + \ldots \beta_k f_k(\mathbf{x}_k) \tag{1}$$

where $\beta_0, \beta_1, \ldots, \beta_k \in \mathbb{Q}$ are the intercept terms and $f(\mathbf{x}) \in \mathbb{R}$ is the regressor predictive value. For the scenario where $f(\mathbf{x})$ is a classification model, we assume a step function over the additive prediction, i.e., $f(\mathbf{x}) := \text{step}(\beta_0 + \beta_1 f_1(\mathbf{x}_1) + \beta_2 f_2(\mathbf{x}_2) + \ldots \beta_k f_k(\mathbf{x}_k))$ where we define the step function as $\text{step}(\mathbf{z}) = 1$ if $\mathbf{z} \geq 0$, and $\text{step}(\mathbf{z}) = 0$ otherwise.

# 3 Dimensions of the Complexity Analysis

As previously mentioned, our detailed complexity analysis spans three distinct dimensions: (i) the input domain, (ii) the types of component models utilized within GAMs, and (iii) the explanation types. In this section, we elaborate on each of these dimensions.

## 3.1 Dimension 1: The input domain

We study the following cases for the domain $\mathcal{X}_i$ of feature $i \in \{1, \ldots n\}$: (i) when the domain is *enumerable discrete*, meaning each value $\mathbf{x}_i$ is selected from a given set $\mathcal{K} \subseteq \mathbb{Q}$ of a constant number $|\mathcal{K}|$ of possible assignments. Intuitively, these are scenarios where each input is limited to a predefined set of values, which is often relevant for cases such as tabular data or certain language tasks; (ii) the *discrete* setting, or the *general discrete* setting, where $\mathcal{X}_i$ represents any discrete input within a range defined by maximum and minimum values expressed in binary with $q$ bits. This intuitively allows a broader range of discrete assignments for each input, while still operating within a discrete domain. (iii) The *continuous* setting, where $\mathcal{X}_i \subseteq \mathbb{R}$, which ensures guarantees across the entire infinite spectrum of the specified domain. Both discrete and continuous domains are widely used across various ML tasks, including those in vision and language tasks [46, 52, 36]. We note that general discrete settings naturally encompass enumerable ones and are thus always at least as computationally challenging, whereas continuous settings do not enable a direct comparison to either and can make problems either harder or easier to solve [68, 103, 10].

## 3.2 Dimension 2: The component model type

We consider a diverse range of component models that can be integrated into a GAM, spanning various levels of interpretability and reflecting commonly used approaches in the literature: (i) The classic *spline*-based models [51], where each component function is piecewise polynomial. In practice, these polynomials are typically restricted to a degree of at most three, yielding the widely used *cubic splines* [51], which will be our primary focus (though our results can often be extended to higher-order splines — see Appendix B). To distinguish these GAMs from the non-smooth variants, we refer to GAMs with spline-based components as *Smooth GAMs*. (ii) *Neural Additive Models (NAMs)* [3, 30, 110, 66, 101], in which each component is a neural network with ReLU activations. (iii) *Explainable Boosting Machines (EBMs)* [37, 107, 89], where each component function is a boosted tree ensemble. We present the complete formal definitions of all model types in Appendix B.

## 3.3 Dimension 3: The explanation type

To explore various dimensions of GAM interpretability, we examine a broad range of widely used explanation types drawn from the existing literature. Building upon prior computational complexity frameworks for evaluating explanations in ML models [16, 6, 18], we formalize each type of explanation as an *explainability query*. Such a query takes a model $f$ and $\mathbf{x}$ as inputs and aims to address specific questions, offering a meaningful interpretation of the prediction $f(\mathbf{x})$.

**Sufficient reason Feature Selection.** We consider the common *sufficiency* criterion for feature selection, used in popular explainability methods [93, 35, 58]. A *sufficient reason* is a subset of input features, $S \subseteq [n]$, such that fixing the features in $S$ to their values in $\mathbf{x} \in \mathbb{F}$ guarantees the prediction remains $f(\mathbf{x})$, regardless of the assignment to $\overline{S}$. We write $(\mathbf{x}_S; \mathbf{z}_{\overline{S}})$ for an input where $S$ takes values from $\mathbf{x}$ and $\overline{S}$ from $\mathbf{z}$. Formally, $S$ is a sufficient reason for $\langle f, \mathbf{x} \rangle$ iff for all $\mathbf{z} \in \mathbb{F}$: $f(\mathbf{x}_S; \mathbf{z}_{\overline{S}}) = f(\mathbf{x})$.

A common assumption in the literature is that smaller sufficient reasons (those with smaller $|S|$) are more useful [93, 35, 58]. This motivates the search for *cardinally minimal sufficient reasons*, also called *minimum sufficient reasons*, and leads to our first explainability query:

---

**MSR (Minimum Sufficient Reason)**:
**Input**: Model $f$, input $\mathbf{x}$, and $d \in \mathbb{N}$
**Output**: *Yes* if there exists some $S \subseteq [n]$ such that $S$ is a sufficient reason with respect to $\langle f, \mathbf{x} \rangle$ and $|S| \leq d$, and *No* otherwise.

---

To deepen our understanding of the complexity of sufficient reasons, we also examine two common related explainability queries that refine the MSR query [16, 18]: (i) *Check-Sufficient-Reason* (CSR),

which checks if a given subset $S$ is a sufficient reason; (ii) *Count-Completions* (CC), a generalized version of CSR that measures the fraction of completions preserving the prediction, capturing the *probability* of maintaining the classification. Full formalizations appear in Appendix C.

**Contrastive Explanations.** An alternative way to interpret models is by identifying subsets of features that, when changed, could alter the model's prediction [45, 50]. We define s subset $S \subseteq [n]$ as *contrastive* if modifying its values can change the classification $f(\mathbf{x})$, i.e., there exists $\mathbf{z} \in \mathbb{F}$ such that $f(\mathbf{x}_{\bar{S}}; \mathbf{z}_S) \neq f(\mathbf{x})$. As with sufficient reasons, smaller contrastive subsets are typically assumed to be more meaningful [45, 50, 57, 86], motivating a focus on *cardinally-minimal contrastive reasons*:

---

**MCR (Minimum Change Required)**:
**Input**: Model $f$, input $\mathbf{x}$, and $d \in \mathbb{N}$.
**Output**: *Yes*, if there exists some contrastive reason $S$ such that $|S| \leq d$ for $f(\mathbf{x})$, and *No* otherwise.

---

**Shapley Values.** In the additive attribution setting, each feature $i \in [n]$ is assigned an importance weight $\phi_i$. A common method for assigning these weights is the *Shapley value* attribution index [76]:

$$\phi_i(f, \mathbf{x}) := \sum_{S \subseteq [n] \setminus \{i\}} \frac{|S|!(n - |S| - 1)!}{n!} (v(S \cup \{i\}) - v(S)) \tag{2}$$

We use the standard *conditional expectation* value function $v(S) := \mathbb{E}_{\mathbf{z} \sim \mathcal{D}_p}[f(\mathbf{z}) \mid \mathbf{z}_S = \mathbf{x}_S]$ [100, 76]. Moreover, as is standard in both complexity analyses [8, 102] and practical SHAP methods like KernelSHAP [76], we assume feature independence. See Appendix C for a full formalization.

---

**SHAP (Shapley Additive Explanation)**:
**Input**: Model $f$, input $\mathbf{x}$, and $i \in [n]$
**Output**: The shapley value $\phi_i(f, \mathbf{x})$.

---

**Feature redundancy.** For our final explanation form, we study the complexity of determining whether a feature $i$ is redundant with respect to a model $f$, following the notion of redundancy explored in prior work [6, 18, 53, 43]. Unlike the previous local explanation forms, this one is global — it checks if $i$ is redundant for all $\mathbf{x} \in \mathbb{F}$. Formally, $i$ is *redundant* if for all $\mathbf{x}, \mathbf{z} \in \mathbb{F}$, we have $f(\mathbf{x}) = f(\mathbf{x}_{[n] \setminus \{i\}}; \mathbf{z}_{\{i\}})$.

---

**FR (Feature Redundancy)**:
**Input**: Model $f$, and integer $i$.
**Output**: *Yes*, if $i$ is redundant with respect to $f$, and *No* otherwise.

---

## 4 Main Complexity Results

We examine the complexity across all analyzed settings — spanning input domains (enumerable discrete, discrete, continuous), model types (NAMs, EBMs, Smooth GAMs), and explanation forms (CSR, MSR, MCR, FR, CC, SHAP). We distinguish between SHAP for regression and classification (SHAP-R vs. SHAP-C), a complexity split, which, as we explain in detail later, is unique to SHAP. For each setting, we provide a novel complexity proof, beginning with a summary in Table 1.

The results in Table 1 identify when explanations can be computed efficiently (e.g., in polynomial or pseudo-polynomial time) and when they are computationally intractable (e.g., NP-Hard). A central finding is the *significant variation in complexity*, influenced by factors such as the input domain, the type of component model, and whether the task is regression or classification. In Section 5, we analyze this diverse landscape in depth and compare these complexity results to existing complexity results for non-additive models like neural networks and tree ensembles. In Section 6, we will outline the proofs underlying these complexity results.

Table 1: A summary of complexity results across all component-model types, input domains, and explanations. Bold font emphasizes the most tractable complexity classes for each explanation.

| Input Space | Component Type | CSR, MSR | MCR | FR | CC, SHAP-C | SHAP-R |
|---|---|---|---|---|---|---|
| Enumerable Discrete | Any | **PTIME** | **PTIME** | coNP-C | #P-C **(Pseudo-P)** | **PTIME** |
| Discrete | Smooth GAMs | **PTIME** | **PTIME** | coNP-C | #P-C | **PTIME** |
| | NAMs, EBMs | coNP-H | NP-C | coNP-C | #P-C | #P-C |
| Continuous | Smooth GAMs | **PTIME** | **PTIME** | **PTIME** | #P-C | **PTIME** |
| | NAMs, EBMs | coNP-H | NP-C | coNP-C | #P-C | #P-C |

# 5 A Computational Interpretability Hierarchy for GAMs

Our results uncover a rich and varied landscape of complexity outcomes across different settings, which we analyze in this section. To present these complexity distinctions more elegantly, we adopt the notation from [16], indicating when one model is strictly *more computationally interpretable* (c-interpretable) than another — that is, when it belongs to a strictly lower complexity class.

**Definition 1.** *Let $\mathcal{C}_1$ and $\mathcal{C}_2$ be two classes of models and let $Q$ be an explainability query for which $Q(\mathcal{C}_1)$ is in complexity class $\mathcal{K}_1$ and $Q(\mathcal{C}_2)$ is in complexity class $\mathcal{K}_2$. We say that $\mathcal{C}_1$ is strictly more c-interpretable than $\mathcal{C}_2$ with respect to $Q$ iff $Q(\mathcal{C}_2)$ is hard for the complexity class $\mathcal{K}_2$ and $\mathcal{K}_1 \subsetneq \mathcal{K}_2$.*

## 5.1 The Sensitivity of Complexity to the Input Domain, in Contrast to Other ML Models

Our results reveal significant exponential complexity distinctions in explaining GAMs based on the input domain — a surprising finding, as such distinctions do not appear in other popular ML models (e.g., decision trees, tree ensembles, neural networks; see Appendix D for details). While explanations over enumerable discrete domains are often computable in polynomial time, they become intractable in general discrete or continuous settings. The only exception is the feature redundancy query.

**Theorem 1.** *GAMs over enumerable discrete settings are strictly more c-interpretable than GAMs over general discrete or continuous settings with respect to CSR, MCR, MSR, CC and SHAP.*

The intuition is that in an enumerable discrete setting, the input space can be explicitly traversed to evaluate each component $f_i$, allowing computations such as minimum or maximum component output computation — key for sufficient or contrastive reason queries — as well as expected value computation, which is essential for SHAP. However, in the general discrete or continuous setting, such iteration is infeasible, making these computations intractable.

However, interestingly, we discover that for one of the queries we analyzed — the Feature Redundancy (FR) query, which seeks to determine whether a feature does not contribute to a model's prediction — this type of explanation is actually strictly easier to obtain in the continuous setting compared to the discrete or even the enumerable discrete settings:

**Theorem 2.** *GAMs over continuous input settings are strictly more c-interpretable than GAMs over enumerable discrete or general discrete input settings with respect to FR.*

We highlight that these complexity gaps, tied to input domains, are *unique to additive models*. For other model types — decision trees, neural networks, and tree ensembles — no such gaps appear (see Appendix D for an elaborate discussion):

**Observation 1.** *While there exist strict complexity gaps for the CSR, MCR, CC, SHAP, and FR queries between the different input setting configurations for GAMs, such complexity gaps do not hold for other ML models such as decision trees, tree ensembles, and neural networks.*

## 5.2 The Significance of the Component Models on Complexity Depends on the Input Domain

We show that while computing some explanations for certain GAMs with more interpretable component models (such as splines) can be done efficiently, these same tasks may sometimes become intractable for GAMs consisting of uninterpretable component models (e.g., neural networks or boosted trees). We show that this phenomenon holds *only* in the continuous and discrete input spaces:

**Theorem 3.** *GAMs over continuous or discrete inputs composed of splines (Smooth GAMs) are strictly more c-interpretable than GAMs with neural networks or tree ensembles (NAMs, EBMs).*

However, we surprisingly demonstrate that this distinction between model components does not always hold and is *dependent on the input domain*. Specifically, we show that this gap emerges only in the continuous and discrete input settings but not in the enumerable discrete setting. The intuition behind this result is that the enumerability of the input space mitigates the complexity originating from the component model itself, unlike in the discrete or continuous domains:

**Observation 2.** *While there exist strict complexity distinctions between component models when computing explanations for GAMs in the discrete and continuous input domain, such complexity distinctions do not hold under the enumerable discrete input domain.*

We also note that the result shown in this subsection, together with our findings in Subsection 5.1 on input domain sensitivity, offers a complementary perspective to empirical work such as [39]. These studies demonstrate that both the choice of component model and the form of data representation (i.e., input representation) substantially influence the learned functional structure of GAMs. Since differing functional forms naturally correspond to different levels of computational complexity, this aligns with our observation that the complexity of generating explanations is highly sensitive to these modeling choices.

### 5.3  Complexity Distinctions Between Regression and Classification, but Only for SHAP

In the context of SHAP explanations, we reveal an intriguing complexity distinction between classification and regression GAMs. Specifically, we show that while SHAP explanations for regression tasks on Smooth GAMs (or general GAMs with enumerable discrete inputs) can be computed in polynomial time, the same problem becomes computationally intractable (#P-hard) for classification tasks. The intuition underlying this distinction is that the classification scenario introduces an additional "step" function, whose non-linearity violates the linearity axiom essential for establishing tractability in the regression setting. This result motivates the following:

**Theorem 4.** *Regression Smooth GAMs and regression GAMs with enumerable inputs are strictly more c-interpretable than classification Smooth GAMs and classification GAMs with enumerable inputs with respect to SHAP.*

We note that while SHAP allows for a direct comparison between regression and classification tasks, adapting our other explanation types — originally defined for classification — to regression requires redefining them (as in [109, 61]) to assess whether values lie within a specified $\delta$-range. From a complexity standpoint, these behave similarly to classification cases and do not produce the same distinctions observed with SHAP. We elaborate on this in Appendix D.

### 5.4  Additive vs. Non-Additive Complexity Depends on the Input Domain and Explanation

As noted in the introduction, the ML community often assumes that increasing a model's additivity improves interpretability. For instance, NAMs [3, 92] and EBMs [37] are generally seen as more interpretable than standard neural networks and boosted trees, respectively. It thus seems natural to expect that explanations for additive models (e.g., NAMs) are easier to compute than their non-additive counterparts. Our findings support this, showing a clear complexity gap between black-box models and their additive counterparts. However, importantly, this gap is *not* universal — it strongly depends on the explanation type and the input domain. Table 2 summarizes prior complexity results for neural networks and tree ensembles, alongside our new results for NAMs and EBMs.

Particularly, we show that in the enumerable discrete input domain, there exist many complexity gaps between the non-additive models — neural networks and tree ensembles, and their additive counterparts (NAMs and EBMs). However, in the discrete and continuous input settings, while such a complexity gap is apparent in the minimum-sufficient-reason (MSR) query, it is not apparent in the other queries. These observations lead us to the following corollary:

**Theorem 5.** *NAMs and EBMs over enumerable discrete input settings are strictly more c-interpretable than neural networks and tree ensembles with respect to CSR, MSR, MCR, and SHAP for regression. However, NAMs and EBMs over general discrete or continuous input settings are strictly more c-interpretable than tree ensembles and neural networks with respect to MSR.*

Table 2: Complexity distinctions in explaining *non-additive* black-box models — tree ensembles (TEs) or neural networks (NNs), compared to their *additive counterparts* — neural additive models (NAMs) and explainable boosting machines (EBMs). We emphasize in bold font cases where a strict complexity separation exists between an additive and a non-additive configuration.

| | Enumerable Discrete | | Discrete, Continuous | |
| | EBMs, NAMs | TEs, NNs | EBMs, NAMs | TEs, NNs |
| --- | --- | --- | --- | --- |
| CSR | **PTIME** | **coNP-C** | coNP-C | coNP-C |
| MSR | **PTIME** | $\mathbf{\Sigma_2^P}$**-C** | **coNP-H, in** $\Delta_2^P$ | $\mathbf{\Sigma_2^P}$**-C** |
| MCR | **PTIME** | **NP-C** | NP-C | NP-C |
| FR | coNP-C | coNP-C | coNP-C | coNP-C |
| CC, SHAP-C | #P-C **(Pseudo-P)** | #P-C **(No Pseudo-P)** | #P-C | #P-C |
| SHAP-R | **PTIME** | **#P-C** | #P-C | #P-C |

## 6 Proof Outline of Complexity Results

We begin by summarizing the tractable results — explanations that can be computed in polynomial time — in Subsection 6.1. Next, in Subsection 6.2, we discuss a relaxed setting where tractability is preserved under unary-encoded weights, referred to as pseudo-polynomial time. Finally, Subsection 6.3 covers the intractable cases, such as NP-hardness and #P-hardness results.

### 6.1 Polynomial-Time Explanation Computations

**Sufficient and contrastive reasons.** We begin by analyzing the polynomial-time complexity of computing sufficient and contrastive reason queries (CSR, MSR, MCR). A key observation is that by identifying the minimal and maximal values each component $\beta_i \cdot f_i(\mathbf{x}_i)$ can take, we can rank features by their "importance", measured by how much altering each feature deviates the prediction. A greedy algorithm, akin to those in [16, 78], then selects features by importance until a minimal sufficient or contrastive reason is found. CSR is simpler, requiring only a sufficiency check for a given subset. Thus, the main computational challenge lies in efficiently computing the min/max values of each $f_i(\mathbf{x}_i)$. For enumerable discrete inputs, this can be done by enumeration; for smooth GAMs, these values can be computed analytically, even with general discrete or continuous inputs.

**Proposition 1.** *Given any GAM over an enumerable discrete domain, or a Smooth GAM over a discrete or continuous domain, the CSR, MCR, and MSR queries are solvable in polynomial time.*

**Feature redundancy for Smooth GAMs in continuous settings.** Unlike the discrete or enumerable discrete settings, where identifying strict feature redundancies is intractable (see Subsection 6.3), in continuous domains we show that a feature is strictly redundant if and only if its component $\beta_i \cdot f_i(\mathbf{x}_i)$ is identically zero — i.e., either $\beta_i = 0$ or $f_i(\mathbf{x}_i) = 0$ for all $\mathbf{x}_i$. For Smooth GAMs, this can be checked efficiently in polynomial time. This leads to the following proposition:

**Proposition 2.** *Given a Smooth GAM $f$ over a continuous input space, the FR query can be obtained in polynomial time.*

**SHAP explanations in regression settings.** When computing SHAP, polynomial-time computation is feasible only for GAMs in *regression* tasks (classification intractability is discussed in Subsection 6.3). Leveraging the linearity axiom of Shapley values [76], we reduce the computation to evaluating the computations of $\beta_i \cdot \mathbb{E}_{\mathbf{z} \sim \mathcal{D}_p}[f_i(\mathbf{x}_i)]$. When the model is either discretely enumerable or supports efficient integration (e.g., in Smooth GAMs), these expectations can be computed in polynomial time.

**Proposition 3.** *Given any regression GAM with enumerable discrete inputs, or given a regression Smooth GAM with discrete or continuous inputs, then the SHAP query is solvable in polynomial time.*

### 6.2 Pseudo Polynomial-Time Explanation Computations

Table 1 shows that both CC and SHAP-C queries are intractable. However, if the GAM's internal encoding and input is given in *unary* (rather than *binary*) and the GAM has a polynomially bounded output, they become computable in polynomial time. This *pseudo-polynomial* setting enables efficient computation when the encoding precision is bounded, using dynamic programming similar to [16],

and detailed in our proofs. It applies to enumerable discrete domains (all component types), where enumeration makes $\mathbb{E}_{\mathbf{z} \sim \mathcal{D}_p}[f(\mathbf{x}_i)]$ tractable.

**Proposition 4.** *Given any unary-encoded GAM over a discrete, enumerable input domain with polynomially bounded output, the CC and SHAP-C queries admit a pseudo-polynomial-time algorithm.*

### 6.3 Intractable Explanation Computations

**Sufficient and contrastive reasons.** As shown in Subsection 6.1, finding a cardinally minimal sufficient or contrastive reason in GAMs requires computing the minimal and maximal values of each $f_i(\mathbf{x}_i)$ component. We show this is intractable for neural networks and tree ensembles, as their exponential search space makes the problem computationally hard. The main difficulty is proving hardness even for single input-output instances, where lower expressivity demands more intricate constructions. We overcome this by reducing the instance to an equivalent $n$-dimensional input, mapping continuous or general discrete regions to their relevant counterparts, yielding the following:

**Proposition 5.** *Let there be a NAM or an EBM over a discrete or continuous input space, then solving the CSR query is coNP-Complete, the MSR query is coNP-Hard, and the MCR query is NP-Complete.*

**Feature redundancy.** In contrast to the continuous case, where redundancy reduces to checking if $\beta_i \cdot f_i(\mathbf{x}_i) = 0$, in discrete settings it requires verifying whether this value falls below the decision boundary — a task that is often computationally intractable:

**Proposition 6.** *Given a NAM, a Smooth GAM or an EBM under an enumerable discrete or general discrete setting, then the FR query is coNP-Complete.*

**Intractability of SHAP and count completions.** Finally, we show that computing the SHAP and CC queries is generally #P-Hard by reducing to the model counting problem [102], implying intractability for EBMs and NAMs. While other settings are also intractable (see Subsection 6.2), their complexity can be mitigated using pseudo-polynomial algorithms:

**Proposition 7.** *Given NAMs and EBMs over discrete or continuous settings, computing SHAP for regression tasks is #P-Complete. Moreover, given* any *GAM over an enumerable discrete, discrete or continuous setting — computing the CC and SHAP queries for classification are #P-Complete.*

## 7 Practical Implications

Although our work is primarily theoretical, it provides several insights with practical relevance for practitioners working with GAMs. First, we identify a broad class of *efficient polynomial-time* algorithms for computing explanations with different guarantees across diverse GAM settings. In fact, all of the tractable algorithms presented in this work (included in the appendix) are not only polynomial-time, but even *linear* in the size of the GAM, underscoring their practical applicability.

Second, we show that techniques commonly used in machine learning practice can also make explanation methods computationally feasible. In particular, we identify two strategies that can transform otherwise hard explanation problems into tractable ones:

1. **Input domain transformations**. We find that the computational complexity of generating explanations is highly sensitive to the structure of the input domain. Small changes in the domain can shift problems from intractable to tractable. This insight motivates the use of *input transformations*, such as discretization, a standard step in many GAM pipelines [37, 89], to reshape the input space in ways that make explanation computation efficient.

2. **Quantization**. Our *pseudo*-polynomial algorithms show that explanation problems that are intractable in full generality become tractable when the GAM encoding is quantized. This suggests a promising practical approach: by quantizing model coefficients, one can enable efficient computation of explanations that would otherwise be infeasible.

## 8 Related Work

Our work builds upon prior research exploring the computational complexity of obtaining various types of explanations for ML models [16, 9, 28, 1, 27, 53, 15, 26, 72, 83, 73, 31, 111, 67]. This area

is closely tied to the subfield of interest known as *formal explainable AI* [79, 24] which focuses on explanations with mathematical guarantees [58, 109, 62, 13, 43, 20, 64, 17, 32], where analyzing the complexity of producing explanations with such guarantees plays a central role [60, 91, 11, 42, 9, 90, 102].

Previous work has mainly examined the complexity of generating explanations for specific ML models, including black-box models like neural networks [2, 16] and interpretable models like decision trees [9, 33, 54]. Closer to our work are studies on linear models [16, 99, 78] — a simplified form of GAMs — typically under binary input-output settings. In contrast, we offer a comprehensive analysis of a broad class of GAMs, covering diverse component models, explanation types, input assumptions, and approximation results. We elaborate further on related works in Appendix A.

Lastly, we note that some terms in our work have sometimes been referred to differently in prior literature. For instance, while the term sufficient reasons is widely used [16, 23, 43], it is also referred to as abductive explanations [59]. Moreover, subset-minimal sufficient reasons are closely related — though not identical — to prime implicants in Boolean formulas [44, 43, 97].Similarly, the CC query corresponds to the $\delta$-relevant set [104, 62, 99], which checks if the completion count exceeds a given threshold $\delta$.

# 9 Limitations and Future Work

As in prior work on the computational complexity of obtaining explanations, our analysis focuses on *specific* explanation definitions and component model types, though many other settings could certainly be explored. Nevertheless, we believe that our work provides a broad perspective across a wide array of popular explanation methods, component types, and input settings, highlighting various novel aspects of the explainability landscape for GAMs and establishing a strong foundation for exploring additional settings in future research.

More concretely, our results open the door to a wide range of future practical implementations of our algorithms (see Section 9). From a theoretical perspective, our analysis can be extended in several directions. A first promising avenue is to better understand how *high-order interactions*, which are widely used in GAMs [38, 75, 70, 48], affect the complexity of generating explanations, and how this varies across explanation types and component models. Another interesting direction is to study how structural assumptions over the GAM influence complexity. For instance, one may consider the effect of *concurvity* [71, 98]. In the extreme case of high concurvity, where smooth components are strongly correlated and behave almost like a single spline, the GAM may resemble a simpler function class, potentially making some explanation problems easier. Conversely, in the case of low concurvity, especially for explanation types that rely on *interactions*, the high separability between components may also offer computational advantages. Finally, for intractable settings, one can attempt to leverage automated reasoning tools, such as neural network verifiers [108, 106] or MILP/MaxSAT solvers for tree ensembles [56, 41] to obtain explanations. Moreover, while we propose two central strategies for circumventing intractability, *input transformations* and *quantization* (see Section 9), exploring additional circumvention approaches such as approximation guarantees, probabilistic or statistical relaxations, and PAC-based guarantees remains an important direction for future research.

# 10 Conclusion

We present a theoretical framework for assessing the computational complexity involved in obtaining various forms of explanations for different types of GAMs. Our work uncovers a nuanced spectrum of results, demonstrating that the complexity of obtaining explanations for GAMs is significantly influenced by (i) the input space domain (enumerable discrete, discrete, and continuous), (ii) the type of explanation (sufficient, contrastive, shapley value explanations, etc.), (iii) the underlying component model (e.g., neural networks, boosted trees, splines, etc.), (iv) and the distinction between classification and regression settings. Our findings reveal a broad range of unexpected and substantial complexity variations, driven by surprising factors — such as the input domain — which are not apparent in other ML models. We believe that our work lays the foundation for a wide range of implementations by enabling the development of tractable algorithms for computing explanations for GAMs in diverse settings. At the same time, it advances the theoretical understanding of these models by identifying when such explanations are computationally feasible and when they are not.

## Acknowledgments

This work was partially funded by the European Union (ERC, VeriDeL, 101112713). Views and opinions expressed are however those of the author(s) only and do not necessarily reflect those of the European Union or the European Research Council Executive Agency. Neither the European Union nor the granting authority can be held responsible for them. This research was additionally supported by a grant from the Israeli Science Foundation (grant number 558/24).

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

# Appendix

The appendix provides formal definitions and proofs referenced throughout the paper.

**Appendix A** discusses extended related work.
**Appendix B** formalizes the GAMs that were studied in this work.
**Appendix C** provides additional details regarding the explainability queries analyzed in this work.
**Appendix D** contains possible extensions of our framework and its relationship to previous results.
**Appendix E** contains the proof of Proposition 1.
**Appendix F** contains the proof of Proposition 2.
**Appendix G** contains the proof of Proposition 3.
**Appendix H** contains the proof of Proposition 4.
**Appendix I** contains the proof of Proposition 5.
**Appendix J** contains the proof of Proposition 6.
**Appendix K** contains the proof of Proposition 7.

## A   Extended Related Work

In this section, we present a more technical overview of related work relevant to our framework. While all explanation queries we examine have been explored in prior studies, they were analyzed in the context of different models [16, 18, 1, 53, 54, 42, 102, 82, 49, 5, 65, 34, 81]. Notably, some results for neural networks [16, 1, 22] and tree ensembles [91, 54, 14, 19] are somewhat related to our setting, as NAMs and EBMs are instances of GAMs that incorporate neural networks or tree ensembles as base models. However, as demonstrated in our work (e.g., Table 2), the shift from additive to non-additive structures leads to fundamentally different complexity behaviors, necessitating entirely distinct proof techniques for both membership and hardness results.

Another line of related work involves studies on the computational complexity of generating specific types of explanations for linear models (e.g., [16, 99, 78, 102]). Some of the constructions used in our proofs draw upon ideas introduced in these works, and we reference them where appropriate — for instance, the use of sorting algorithms in the MSR/MCR queries and dynamic programming techniques for the CC and SHAP queries. That said, our study is the first to explore these explanations within the context of GAMs, while also addressing a substantially broader and more diverse set of input settings, component models, and explanation types — requiring us to tackle considerably more complex instances. For instance, proving hardness for various explanation queries over general discrete and continuous domains in models like NAMs and EBMs required novel constructions that differ substantially from those applicable to linear models.

## B   Generalized Additive Models

In this appendix, we outline the different generalized additive models (GAMs) analyzed in this work. Our focus will be on GAMs used for either regression or classification tasks. For simplicity, we assume the GAM produces a single output: a value in $\mathbb{R}$ for regression tasks or a single class from $\{0, 1\}$ for classification tasks. However, our results can be extended to handle multiple regression outputs or multi-label classification. Specifically, we formalize a GAM for a regression task as follows:

$$f(\mathbf{x}) := \beta_0 + \beta_1 f_1(\mathbf{x}_1) + \beta_2 f_2(\mathbf{x}_2) + \ldots + \beta_k f_k(\mathbf{x}_k) \tag{3}$$

where $f_1, \ldots, f_k$ represent uncorrelated functions $f_i : \mathcal{X}_i \to \mathbb{R}$, and $\beta_0, \beta_1, \beta_2, \ldots, \beta_k \in \mathbb{Q}$ denote the intercept terms obtained during a training process. For classification tasks, we define similarly:

$$f(\mathbf{x}) := \text{step}\big(\beta_0 + \beta_1 f_1(\mathbf{x}_1) + \beta_2 f_2(\mathbf{x}_2) + \ldots + \beta_k f_k(\mathbf{x}_k)\big) \tag{4}$$

where we define the step function as $\text{step}(\mathbf{z}) = 1$ if $\mathbf{z} \geq 0$, and $\text{step}(\mathbf{z}) = 0$ otherwise.

## B.1 Component-Model Types

In this subsection, we define the component-model types discussed throughout the paper: (i) spline-based components within Smooth GAMs, (ii) neural network components within neural additive models (NAMs), and (iii) boosted tree components within explainable boosting machines (EBMs) .

**Spline components for Smooth GAMs.** Here, we define each $f_i$ as the standard piecewise spline function, corresponding to each component of the GAM [51]. Specifically, the domain $\mathcal{X}_i$ for each feature $\mathbf{x}_i$ is divided into intervals $[a_1, a_2), [a_2, a_3), \ldots, [a_{d-1}, a_d]$, and a polynomial is defined for each interval $\mathcal{W}$. For $w \in \mathcal{W}$, we write the polynomial as $g(w) := \alpha_r w^r + \alpha_{r-1} w^{r-1} + \ldots + \alpha_1 w + \alpha_0$. The function $f_i(\mathbf{x}_i)$ is defined over the domain $\mathcal{X}_i$ by assigning the value of the corresponding polynomial $g$ for each interval. Typically, the polynomials used in GAMs are of degree at most 3 (referred to as *cubic* splines), which is the assumption we adopt here. We note, however, that many of our results also extend to higher-order polynomial splines. In particular, all of our *hardness* results remain valid in these cases, as such splines are at least as complex as cubic splines. As for the *membership* results, our findings for enumerable discrete settings still apply, since they rely only on the assumption that inference over each component can be performed in polynomial time — a condition satisfied by splines of any degree.

We also observe that in the simple case where each $f_i(\mathbf{x}_i)$ is the identity function — which is trivially captured by splines as well as our other models — the overall GAM $f(\mathbf{x})$ effectively reduces to a standard linear model. This linear model can either be a regressor $f(\mathbf{x}) := \beta_0 + \beta_1 \mathbf{x}_1 + \ldots + \beta_k \mathbf{x}_k$ or a classification model $f(\mathbf{x}) := \text{step}(\beta_0 + \beta_1 \mathbf{x}_1 + \ldots + \beta_k \mathbf{x}_k)$.

**Neural network components for Neural Additive Models (NAMs).** We denote a neural network $f$, consisting of $t - 1$ *hidden layers* ($g^j$ where $j$ ranges from 1 to $t - 1$) and a single output layer ($g^t$). The layers are defined recursively — each layer $g^{(j)}$ is computed by applying the activation function $\sigma^{(j)}$ to the linear combination of the outputs from the previous layer $g^{(j-1)}$, the corresponding weight matrix $W^{(j)}$, and the bias vector $b^{(j)}$. This is represented as $g^{(j)} := \sigma^{(j)}(g^{(j-1)} W^{(j)} + b^{(j)})$ for each $j$ in $1, \ldots, t$. The model includes $t$ weight matrices ($W^{(1)}, \ldots, W^{(t)}$), $t$ bias vectors ($b^{(1)}, \ldots, b^{(t)}$), and $t$ activation functions ($\sigma^{(1)}, \ldots, \sigma^{(t)}$).

In this neural network, the function $f$ is defined to output $f := g^{(t)}$. The initial input layer $g^{(0)}$ is denoted by $\mathbf{x}$, which serves as the model's input. The dimensions of the biases and weight matrices are specified by the sequence of positive integers $d_0, \ldots, d_t$. We specifically consider weights and biases that are rational numbers, represented as $W^{(j)} \in \mathbb{Q}^{d_{j-1} \times d_j}$ and $b^{(j)} \in \mathbb{Q}^{d_j}$, which are parameters optimized during training. Since the model is a binary classifier for indices $1, \ldots, n$, it follows that $d_0 = 1$ (since the input of each component includes only one feature) and $d_t = 1$. The primary activation function $\sigma^{(i)}$ that we consider is the *ReLU* activation function, defined as $ReLU(x) = \max(0, x)$. For classification tasks, we assume the output layer includes a *sigmoid* function. However, since our focus is on the post-hoc interpretation of the corresponding model, we equivalently assume, in the case of classification, the existence of a step function for the final layer activation.

**Boosted tree ensemble components for Explainable Boosting Machines (EBMs)** In this scenario, we assume that each individual model $f_i(\mathbf{x}_i)$ is a boosted tree ensemble. We begin by formalizing decision trees, followed by an extension to tree ensembles. A decision tree is an acyclic-directed graph that serves as a graphical model for a function $t : \mathcal{X} \to \mathbb{R}$ in regression and $t : \mathcal{X} \to \{0, 1\}$ in classification. This graph represents the given function as follows: (i) Each internal node $v$ is associated with a unique decision rule, assumed to be of the form $\mathbf{x}_i \geq r$ or $\mathbf{x}_i < r$ for any $r \in \mathbb{Q}$, (ii) Every internal node $v$ has exactly two outgoing edges corresponding to the values $\{0, 1\}$ assigned to $v$, where 1 indicates the decision rule is satisfied, and 0 indicates it is not; (iii) Each leaf node is associated with a numerical value in $\mathbb{R}$ (for regression tasks) or with an associated class $j \in [c]$ (for classification tasks). Consequently, assigning a value to the input uniquely determines a path from the root to a leaf in the DT.

Regarding tree ensembles, each $f_i$ consists of an ensemble of $k^i \in \mathbb{N}$ decision trees (defined earlier) $t_1, \ldots, t_{k^i}$, with each tree assigned a weight $\phi_i \in \mathbb{Q}$. In regression tasks, $f_i$ is expressed as a weighted sum of the individual tree outputs: $f_i(\mathbf{x}_i) := \phi_1 t_1(\mathbf{x}_i) + \phi_2 t_2(\mathbf{x}_i) + \ldots + \phi_{k^i} t_{k^i}(\mathbf{x}_i) + \phi_0$. For

classification, the prediction corresponds to the class receiving the highest weighted vote: $f_i(\mathbf{x}_i) := \arg\max_{j \in [c]} \sum_{i=1}^{k^i} \phi_i \cdot \mathbb{I}[t(\mathbf{x}_i) = j]$, where $\mathbb{I}$ is the indicator function.

## C  Additional Query Formalizations

Here, we define the two remaining queries discussed throughout the paper: the *Check-Sufficient-Reason* (*CSR*) and *Count Completions* (*CC*) queries, both previously studied in [16, 18]. We then elaborate on the computational complexity of computing Shapley values, as examined in this work.

The *CSR* query determines whether a given subset $S$ qualifies as a sufficient reason. More formally:

---

**Check Sufficient Reason (CSR):**
**Input**: A model $f$, an instance $\mathbf{x}$, and a subset $S$.
**Output**: *Yes*, if $S$ is a sufficient reason of $\langle f, \mathbf{x} \rangle$, and *No* otherwise.

---

For the *CC* query, we consider a relaxed variant of the CSR query. Rather than checking whether a specific subset is sufficient, it measures the relative proportion of feature completions that preserve the original prediction, assuming the remaining features are independently and uniformly distributed. We begin by defining the *completion count* for a given subset:

$$c(S, f, \mathbf{x}) := \frac{|\{\mathbf{z} \in \mathcal{X}_{\bar{S}} \mid f(\mathbf{x}_S; \mathbf{z}_{\bar{S}}) = f(\mathbf{x})\}|}{|\{\mathbf{z} \in \{0,1\}^{|\bar{S}|}|} \tag{5}$$

Where $\mathcal{X}_{\bar{S}}$ denote the joint domain of the input features $\mathcal{X}_i$ for all $i \in \overline{S}$. The *CC* query is then defined as follows:

---

**CC (Count Completions):**
**Input**: Model $f$, input $\mathbf{x}$, and subset of features $S$.
**Output**: The completion count $c(S, f, \mathbf{x})$.

---

We present here a more detailed formalization of the *Shapley value* attribution method used in the paper. The *Shapley value* is defined as follows:

$$\phi_i(f, \mathbf{x}) := \sum_{S \subseteq [n] \setminus \{i\}} \frac{|S|!(n - |S| - 1)!}{n!} (v(S \cup \{i\}) - v(S)) \tag{6}$$

where $v(S)$ is the *value function*, and we use the common *conditional expectation* value function $v(S) := \mathbb{E}_{\mathbf{z} \sim \mathcal{D}_p}[f(\mathbf{z})|\mathbf{z}_S = \mathbf{x}_S]$ [100, 76]. We follow common conventions in frameworks that assessed the computational complexity of computing Shapley values [8, 102], as well as practical frameworks that compute Shapley values, such as the kernelSHAP method in the SHAP libary [76], and assume that each input feature is independent of all other features. In the discrete settings, every feature $i \in [n]$ is assigned some probability value $[0, 1]$. These are called product distributions in the work of [8] or fully-factorized in the work of [102]. Formally, given some set of discrete values $[k]$, we can describe a probability function $p : [n] \times [k] \to [0, 1]$. For example, $p(2, 7) = \frac{1}{9}$, implies that the probability of feature $i = 2$, to be set to the value $k = 7$ is $\frac{1}{9}$. Then we can define $\mathcal{D}_p$ as an independent distribution over $\mathcal{X}$ iff:

$$\mathbf{Pr}(\mathbf{x}) := \left( \prod_{i \in [n], \mathbf{x}_i = j} p(i, j) \right) \tag{7}$$

The uniform distribution is a special case of $\mathcal{D}_p$, obtained by setting $p(i) := \frac{1}{k}$ for every $i \in [n]$. In the continuous setting, we define a probability function $p(\mathbf{x}_i) \in [0, 1]$ for each feature $\mathbf{x}_i$. Under the feature independence assumption, the joint probability for the continuous input setting is given by $\mathbf{Pr}(\mathbf{x}) := \prod_{i \in [n]} p(\mathbf{x}_i)$. Additionally, we assume that $p(\mathbf{x}_i)$ satisfies certain basic computational properties — specifically, that the sum $\sum_{j \in \mathcal{X}_i} p(i, j)$ can be computed in polynomial time (which we

term poly-summability), or in the continuous case, that integration over $\mathcal{X}_i$ can be done in polynomial time (termed poly-enumerability). These conditions are satisfied by many common distributions, including the uniform distribution, where each input is assigned a fixed probability.

## D   Framework Extensions and Relationship to Previous Results

**Regression, Probabilistic and Multi-Label Classification.** In this segment, we discuss extensions of our framework to broader output configurations, relevant relaxations, and how these settings relate to existing works.

Our definitions can potentially be relaxed to incorporate *probabilistic* notions of sufficiency [104, 62, 9, 105], multi-output classification, or applications within bounded $\epsilon$-ball regions [109, 61, 77]. In particular, the framework could extend beyond binary classification to cover regression or probabilistic classification. For example, in a regression setting where $f : \mathbb{F} \rightarrow \mathbb{R}$, a *sufficient reason* may be defined as a subset $S \subseteq \{1, \ldots, n\}$ of input features such that:

$$\forall \mathbf{z} \in \mathbb{F} \quad ||f(\mathbf{x}_S; \mathbf{z}_{\bar{S}}) - f(\mathbf{x})||_p \leq \delta \tag{8}$$

for a given $\ell_p$-norm and some $\delta \in [0, 1]$. Other notions discussed in our work — such as contrastive explanations and the definition of feature redundancy — can similarly be adapted to this framework. From a computational complexity standpoint, transitioning to regression for these queries does not change the underlying complexity, as the problem effectively reduces to a binary decision: whether the output is above or below some threshold $\delta$. Provided that $\delta$ is not fixed, *hardness* results naturally extend to cover the full domain. Likewise, *membership* results continue to rely on guessing witness assignments within this domain. This stands in sharp contrast to the unique SHAP query, whose behavior — and thus complexity — differs significantly between regression and classification settings, as shown in our work. The same reasoning extends to multi-output classification. In terms of *hardness*, results established for the single-output setting carry over directly. For *membership* results — whether they establish polynomial-time tractability or membership in higher complexity classes — it is important to note that they do not rely on the binary nature of inputs. Instead, they are based on guessing witness assignments and verifying conditions such as $f(\mathbf{x}_S; \mathbf{z}_{\bar{S}}) = f(\mathbf{x})$ or $f(\mathbf{x}_S; \mathbf{z}_{\bar{S}}) \neq f(\mathbf{x})$, which apply similarly in both single- and multi-output configurations.

**The input setting impact on other ML models.** In the main paper, we noted that while the input setting plays a significant role in shaping the computational complexity of explanation tasks in GAMs, prior studies on other ML models — specifically decision trees, neural networks, and tree ensembles — have not demonstrated a similar complexity separation based on the input domain. In this section, we provide a detailed justification for this claim, explaining why existing results suggest that such separations do not occur for these models. We begin with the simpler case of decision trees and tree ensembles. Intuitively, transitioning from a discrete to a continuous input domain does not alter the fundamental behavior of these models, as they compute piecewise constant functions (or an ensemble of such functions) and their split rules do not depend on specific input configurations. More concretely, prior work has shown that the complexity results for sufficient and contrastive explanations extend to continuous domains (see, e.g.,[54]), and similar extensions hold for counting-based explanations such as SHAP (e.g.,[55]). Lastly, since tree ensembles are simply aggregations of $k$ decision trees, the same proofs naturally extend to them as well.

In the context of neural networks, hardness results are already established even for Boolean input settings — this holds for the CSR, MCR, MSR, and CC queries [16], the FR query [18], and the SHAP query [102]. These results directly imply hardness in the enumerable discrete setting and, by extension, in the general discrete setting as well. Furthermore, [95] (building on [69]) demonstrated that satisfiability instances over binary inputs can be reduced to *continuous* input cases by introducing additional activation gadgets. This reduction, in turn, implies that the same hardness results hold for all these queries in the continuous setting.

Regarding membership results, prior work has already established that these results hold for binary inputs [16, 18, 102]. The same membership arguments — typically based on guessing a witness assignment — extend naturally to enumerable and general discrete domains, where the guess involves a $q$-bit vector instead of a binary input. Additionally, [95] demonstrate that verifying satisfiability queries for MLPs with ReLU activations over continuous inputs is in NP. The key idea is to guess the activation pattern of each ReLU and then solve a corresponding linear program, which is solvable in

polynomial time. The *CSR* query (with $S = \emptyset$) effectively negates a satisfiability query, implying that *CSR* remains coNP-complete for continuous MLPs. Recall that the *MSR* query for MLPs is known to be $\Sigma_2^P$-complete [16], due to its use of a coNP oracle to check whether a subset is sufficient (i.e., solving the *CSR* query). Since *CSR* extends to continuous domains, the same reasoning carries over to *MSR*. Similar logic applies to the *CC* and *SHAP* queries, which are counting analogues of these reductions [16, 102, 7]. Altogether, these results underscore that for neural networks, the complexity class of each query remains stable across input domains, unlike the distinct separations seen in GAMs.

## E    Proof of Proposition 1

**Proposition 1.** *Given any GAM over an enumerable discrete domain, or a Smooth GAM over a discrete or continuous domain, the CSR, MCR, and MSR queries are solvable in polynomial time.*

*Proof.* To start, we present the polynomial-time algorithms applicable to the enumerable discrete setting, which hold for *any* GAM. We then extend our discussion to Smooth GAMs over general discrete and continuous domains.

We divide the proof for the three explainability queries — CSR, MCR, and MSR — for GAMs over enumerable discrete settings into three separate lemmas, each establishing the corresponding complexity result. We note that the proofs across all these settings follow a similar structure to those used for linear models [16, 99, 78], with the main distinction being the need to account for the dependency on computing the minimum and maximum values of each component model $f_i(\mathbf{x}_i)$ to resolve the query.

**Lemma 1.** *The CSR query can be solved for a GAM $f$ over an enumerable discrete input space in polynomial time.*

*Proof.* Let there be some instance $\langle f, \mathbf{x}, S \rangle$ where $f$ is a GAM. We note that given some feature $i$ we can evaluate its maximum negative affect (which we will call, similarly to [16], its "penalty" by fixing it to the (negative) affecting edge of the domain $\mathcal{X}_i$. Formally, we define:

$$p_i := \begin{cases} \min\{\beta_i \cdot f_i(\mathbf{x}_i) \mid \mathbf{x}_i \in \mathcal{X}_i\} & if \ \ f(\mathbf{x}) > 0 \\ \max\{\beta_i \cdot f(\mathbf{x}_i) \mid \mathbf{x}_i \in \mathcal{X}_i\} & 0 \ \ otherwise \end{cases} \tag{9}$$

Since the values in $\mathcal{X}_i$ are enumerable discrete, both $\min\{\beta_i \cdot f_i(\mathbf{x}_i) \mid \mathbf{x}_i \in \mathcal{X}_i\}$ as well as $\max\{\beta_i \cdot f_i(\mathbf{x}_i) \mid \mathbf{x}_i \in \mathcal{X}_i\}$ can be determined in polynomial time by simply iterating over any possible value of $\mathbf{x}_i$. We can then compute all possible $p_i$ values in polynomial time. We can then simply check whether:

$$\text{sign}(\sum_{i \in S} \beta_i \cdot f_i(\mathbf{x}_i) + \sum_{j \in \bar{S}} p_j + \beta_0) \cdot \text{sign}(\sum_{i \in n} \beta_i \cdot f_i(\mathbf{x}_i) + \beta_0) \geq 0 \tag{10}$$

We recall that $(\sum_{i \in n} \beta_i \cdot f_i(\mathbf{x}_i) + \beta_0)$ represents the value of the input $\mathbf{x}$ when passed through $f$ (prior to the step function), and $(\sum_{i \in S} \beta_i \cdot f_i(\mathbf{x}_i) + \sum_{j \in \bar{S}} p_j + \beta_0)$ denotes the value for the input vector $(\mathbf{x}_S; \mathbf{z}_{\bar{S}})$, where this value is maximally distant from the value of $\mathbf{x}$ when passed through $f$ for all $\mathbf{z} \in \mathbb{F}$ (again, prior to the step function). Consequently, the signs of both sums align if and only if both are either positive or negative. This means the sign of the prediction $f(\mathbf{x})$ before the step function matches the sign of $f(\mathbf{x}_S; \mathbf{z}_{\bar{S}})$ before the step function. Since $(\mathbf{x}_S; \mathbf{z}_{\bar{S}})$ represents the vector whose value through $f$ achieves the maximal distance from $\mathbf{x}$ when passed through $f$ (before the step function), the following equality holds if and only if, for *any* possible $\mathbf{z}' \in \mathbb{F}$, the predictions of $f$ over $\mathbf{x}$ and $(\mathbf{x}_S; \mathbf{z}'_{\bar{S}})$ are both positive (or both negative) before the step function. Therefore, after applying the step function, it holds that $\forall \mathbf{z}' \in \mathbb{F}$, $f(\mathbf{x}) = f(\mathbf{x}_S; \mathbf{z}'_{\bar{S}})$, which holds by definition if and only if $S$ is a sufficient reason concerning $\langle f, \mathbf{x} \rangle$.

$\square$

**Lemma 2.** *The MCR query can be solved for a GAM $f$ over an enumerable discrete input space in polynomial time.*

*Proof.* Given an instance $\langle f, \mathbf{x}, k \rangle$, we use the same definition as in Lemma 1 of $p_i$ to denote the "penalty" of each feature $i$ in the GAM. We perform the following algorithm (Alg. 1), which computes

a cardinally minimal contrastive reason. We assume, without loss of generality, that $f(\mathbf{x}) = 1$. The alternate case, where $f(\mathbf{x}) = 0$, can be addressed similarly by reversing Line 4, which will accordingly reverse the steps in the proof.

---

**Algorithm 1** Cardinally Minimal Contrastive Reason Search

**Input** $f, \mathbf{x}$

1: Compute all values $p_i$ for every $i \in [n]$
2: $F \leftarrow \{1, \ldots, n\}$                       ▷ Features for iteration
3: $S \leftarrow \emptyset$                             ▷ The current sufficient reason
4: Sort $F$ in descending order by the value of $v_i := \beta_i \cdot \mathbf{x}_i - p_i$ for each $i$
5: **for each** $i \in F$ **do**
6:      **if** $\text{sign}(\sum_{i \in \bar{S}} \beta_i \cdot f_i(\mathbf{x}_i) + \sum_{j \in S} p_j + \beta_0) \cdot \text{sign}(\sum_{i \in n} \beta_i \cdot f_i(\mathbf{x}_i) + \beta_0) < 0$ **then**
7:          **Break**
8:      **end if**
9:      $S \leftarrow S \cup \{i\}$
10: **end for**
11: **return** $S$                ▷ $S$ is a cardinally minimal contrastive reason

---

We note that the computations of all $p_i$ values (Line 5 in Algorithm 1) can be computed in polynomial time due to our assumption of an enumerable discrete input space. The discrete input space means that for any $\mathbf{x}_i$ we can simply iterate over the entire domain $\mathcal{X}_i$ in polynomial time and hence obtain both $\min\{\beta_i \cdot f_i(\mathbf{x}_i) \mid \mathbf{x}_i \in \mathcal{X}_i\}$ as well as $\max\{\beta_i \cdot f_i(\mathbf{x}_i) \mid \mathbf{x}_i \in \mathcal{X}_i\}$. Hence, it is clear that the algorithm runs in polynomial time.

To establish the correctness of our Lemma, it remains to demonstrate the validity of the algorithm:

**Lemma 3.** *Algorithm 1 obtains a cardinally minimal contrastive reason for $\langle f, \mathbf{x} \rangle$.*

*Proof.* We will first prove that algorithm 1 produces a valid contrastive reason for $\langle f, \mathbf{x} \rangle$. In Lemma 1, we established that if the inequality $\text{sign}\left(\sum_{i \in S} \beta_i \cdot f_i(\mathbf{x}_i) + \sum_{j \in \bar{S}} p_j + \beta_0\right) \cdot \text{sign}\left(\sum_{i \in n} \beta_i \cdot f_i(\mathbf{x}_i) + \beta_0\right) \geq 0$ holds, then $S$ serves as a sufficient reason for $\langle f, \mathbf{x} \rangle$. Similarly, if the relation $\text{sign}\left(\sum_{i \in \bar{S}} \beta_i \cdot f_i(\mathbf{x}_i) + \sum_{j \in S} p_j + \beta_0\right) \cdot \text{sign}\left(\sum_{i \in n} \beta_i \cdot f_i(\mathbf{x}_i) + \beta_0\right) \geq 0$ holds, then $\bar{S}$ is a sufficient reason, which implies by definition that $S$ is *not* a contrastive reason. Therefore, the following equivalence holds: if and only if $\text{sign}\left(\sum_{i \in S} \beta_i \cdot f_i(\mathbf{x}_i) + \sum_{j \in \bar{S}} p_j + \beta_0\right) \cdot \text{sign}\left(\sum_{i \in n} \beta_i \cdot f_i(\mathbf{x}_i) + \beta_0\right) < 0$, then $S$ is a contrastive reason with respect to $\langle f, \mathbf{x} \rangle$.

We will now demonstrate that the generated set $S$ is a cardinally minimal contrastive reason with respect to $\langle f, \mathbf{x} \rangle$. Let $1 \leq \ell \leq n$ represent the last feature added to $S$ in line 10 of algorithm 1. Then, for $S' := S \setminus \{\ell\}$, it follows that: $\text{sign}\left(\sum_{i \in S'} \beta_i \cdot f_i(\mathbf{x}_i) + \sum_{j \in \bar{S}'} p_j + \beta_0\right) \cdot \text{sign}\left(\sum_{i \in n} \beta_i \cdot f_i(\mathbf{x}_i) + \beta_0\right) \geq 0$, implying that $S'$ is *not* a contrastive reason for $\langle f, \mathbf{x} \rangle$.

Recall that $F$ was sorted in descending order according to the values $v_i := \beta_i \cdot f_i(\mathbf{x}_i) - p_i$, and $S'$ therefore represents the subset of size $|S'|$ that maximizes $\sum_{j \in S'}(\beta_j \cdot f_j(\mathbf{x}_j) - p_j)$ and hence also maximizes the difference

$$(\sum_{i \in n} \beta_i \cdot f_i(\mathbf{x}_i) + \beta_0) - (\sum_{i \in \bar{S}'} \beta_i \cdot f_i(\mathbf{x}_i) + \sum_{j \in S'} p_j + \beta_0) = \sum_{j \in \bar{S}'}(\beta_j \cdot f_j(\mathbf{x}_j) - p_j) \quad (11)$$

This implies that any subset $S'' \subseteq [n]$ of size $|S''| \geq |S'|$ satisfies: $\text{sign}\left(\sum_{i \in S''} \beta_i \cdot f_i(\mathbf{x}_i) + \sum_{j \in \bar{S}''} p_j + \beta_0\right) \cdot \text{sign}\left(\sum_{i \in n} \beta_i \cdot f_i(\mathbf{x}_i) + \beta_0\right) \geq 0$, indicating that $S''$ is not a contrastive reason for $\langle f, \mathbf{x} \rangle$. Overall, we have shown that $S$ is a contrastive reason for $\langle f, \mathbf{x} \rangle$, and that any subset $S''$ of size $|S''| \geq |S'|$ is not a contrastive reason for $\langle f, \mathbf{x} \rangle$. By definition, $|S'| = |S| - 1$, so we have established that there is no subset of size $|S| - 1$ or smaller that is a contrastive reason for $\langle f, \mathbf{x} \rangle$. This confirms that $S$ is a cardinally minimal contrastive reason for $\langle f, \mathbf{x} \rangle$.

$\square$

Having established that algorithm 1 produces a cardinally minimal contrastive reason for $S$, solving the MCR query in polynomial time simply requires running the algorithm and verifying if $|S| \geq k$, thereby completing our proof.

$\square$

**Lemma 4.** *The MSR query can be solved for a GAM $f$ over an enumerable discrete input space in polynomial time.*

*Proof.* Given an instance $\langle f, \mathbf{x}, k \rangle$, we will show that the MSR query can be resolved in polynomial time. For a specific feature $i$, we will again (as in Lemma 1) refer to its "penalty" as the maximum negative impact it can exert on the prediction $f(\mathbf{x})$. Formally, we define $p_i := \min\{\beta_i \cdot f_i(\mathbf{x}_i) \mid \mathbf{x}_i \in \mathcal{X}_i\}$ if $f(\mathbf{x}) > 0$, and $p_i := \max\{\beta_i \cdot f_i(\mathbf{x}_i) \mid \beta_i \in \mathcal{X}_i\}$ otherwise. Furthermore, we require a measure for the value of feature $i$ when it is fixed. We define $v_i := \beta_i \cdot f_i(\mathbf{x}_i) - p_i$. Note that when $\sum_{i \in [n]} \beta_i \cdot f_i(\mathbf{x}_i) + \beta_0 < 0$, i.e., when $f(\mathbf{x}) = 0$, a "good score" will be a highly negative one. Thus, we specifically define $v_i := (\beta_i \cdot f_i(\mathbf{x}_i) - p_i) \cdot \text{sign}(\sum_{i \in [n]} \beta_i \cdot f_i(\mathbf{x}_i) + \beta_0)$. However, for simplicity, we will assume without loss of generality that $f(\mathbf{x}) > 0$, and therefore $v_i := \beta_i \cdot f_i(\mathbf{x}_i) - p_i$. Intuitively, $v_i$ reflects the value of feature $i$ when it is held constant rather than allowed to take on any value, thereby influencing the penalty score $p_i$. With these definitions, we are prepared to propose the following algorithm for obtaining a cardinally minimal sufficient reason for the given instance $\langle f, \mathbf{x} \rangle$.

---

**Algorithm 2** Cardinally Minimal Sufficient Reason Search

**Input** $f, \mathbf{x}$

1: Compute all values $v_i, p_i$ for every $i \in [n]$
2: $F \leftarrow \{1, \ldots, n\}$            ▷ Features for iteration
3: $S \leftarrow \emptyset$            ▷ The current sufficient reason
4: Sort $F$ in descending order by the value of $v_i := \beta_i \cdot f_i(\mathbf{x}_i) - p_i$ for each $i$
5: **for each** $i \in F$ **do**
6:      **if** $\text{sign}(\sum_{i \in S} \beta_i \cdot f_i(\mathbf{x}_i) + \sum_{j \in \bar{S}} p_j + \beta_0) \cdot \text{sign}(\sum_{i \in n} \beta_i \cdot f_i(\mathbf{x}_i) + \beta_0) \geq 0$ **then**
7:          **Break**
8:      **end if**
9:      $S \leftarrow S \cup \{i\}$
10: **end for**
11: **return** $S$            ▷ $S$ is a cardinally minimal sufficient reason

---

**Lemma 5.** *Algorithm 2 obtains a cardinally minimal sufficient reason for $\langle f, \boldsymbol{x} \rangle$.*

*Proof.* First let us prove that Algorithm 2 provides a valid sufficient reason. This result is straightforward since we have proven in Lemma 1 that $S$ is a sufficient reason of $\langle f, \mathbf{x} \rangle$ iff the condition $\text{sign}(\sum_{i \in \bar{S}} \beta_i \cdot f_i(\mathbf{x}_i) + \sum_{j \in S} p_j + \beta_0) \cdot \text{sign}(\sum_{i \in n} \beta_i \cdot f_i(\mathbf{x}_i) + \beta_0) \geq 0$ holds.

We will now demonstrate that the generated set $S$ is a cardinally minimal sufficient reason with respect to $\langle f, \mathbf{x} \rangle$. Let $1 \leq \ell \leq n$ represent the last feature added to $S$ in line 10 of algorithm 2. Then, for $S' := S \setminus \{\ell\}$, it follows that: $\text{sign}\left(\sum_{i \in S'} \beta_i \cdot f_i(\mathbf{x}_i) + \sum_{j \in \bar{S}'} p_j + \beta_0\right) \cdot \text{sign}\left(\sum_{i \in n} \beta_i \cdot f_i(\mathbf{x}_i) + \beta_0\right) < 0$, implying that $S'$ is *not* a sufficient reason for $\langle f, \mathbf{x} \rangle$.

Recall that $F$ was sorted according to the values $v_i := \beta_i \cdot f_i(\mathbf{x}_i) - p_i$ in descending order, and $S'$ therefore represents the subset of size $|S'|$ that maximizes $\sum_{j \in S'} (\beta_j \cdot f_j(\mathbf{x}_j) - p_j)$ and hence $\overline{S'}$ represent the subsets that minimizes the difference $\sum_{j \in \bar{S}'} (\beta_j \cdot f_j(\mathbf{x}_j) - p_j)$. This is equivalent to stating that $\overline{S'}$ represents the set of size $|\overline{S'}|$ that minimizes:

$$\left(\sum_{i \in S'} \beta_i \cdot f_i(\mathbf{x}_i) + \sum_{j \in \bar{S}'} p_j + \beta_0\right) - \left(\sum_{i \in n} \beta_i \cdot f_i(\mathbf{x}_i) + \beta_0\right) = \sum_{j \in \bar{S}'} (\beta_j \cdot f_j(\mathbf{x}_j) - p_j) \quad (12)$$

This implies that any subset $S'' \subseteq [n]$ for which $|S''| < |S'|$ and hence also $|\overline{S''}| \geq |\overline{S'}|$ satisfies: $\text{sign}\left(\sum_{i \in S''} \beta_i \cdot f_i(\mathbf{x}_i) + \sum_{j \in \bar{S}''} p_j + \beta_0\right) \cdot \text{sign}\left(\sum_{i \in n} \beta_i \cdot f_i(\mathbf{x}_i) + \beta_0\right) < 0$, indicating that $\overline{S''}$ is a contrastive reason for $\langle f, \mathbf{x} \rangle$, and hence that $S''$ is not a sufficient reason for $\langle f, \mathbf{x} \rangle$. Overall, we

have shown that $S$ is a sufficient reason for $\langle f, \mathbf{x} \rangle$, and that any subset $S''$ of size $|S''| < |S'|$ is not a sufficient reason for $\langle f, \mathbf{x} \rangle$. By definition, $|S'| = |S| - 1$, so we have established that there is no subset of size $|S| - 1$ or smaller that is a sufficient reason for $\langle f, \mathbf{x} \rangle$. This confirms that $S$ is a cardinally minimal sufficient reason for $\langle f, \mathbf{x} \rangle$.

□

Having established that algorithm 2 produces a cardinally minimal sufficient reason for $S$, solving the MSR query in polynomial time simply requires running the algorithm and verifying if $|S| \geq k$, thereby completing our proof.

□

Having established the complexity results for the three explanation types in the enumerable discrete input setting, we now turn our attention to the general discrete or continuous input setting, focusing on the case of Smooth GAMs. In fact, we will show that this claim holds for a broader class of model families, including other tractable components like decision trees:

**Lemma 6.** *Given a Smooth GAM over a general discrete or continuous input setting —- then the CSR, MCR and MSR queries can be obtained in polynomial time.*

*Proof.* We have established in Lemma 1, Lemma 2, and Lemma 4 that, assuming the minimum and maximum viable values for each model component $f_i$ can be determined within the domain $\mathcal{X}_i$, the CSR, MSR, and MCR queries can be computed in polynomial time. We will now demonstrate this for each corresponding component model.

First, consider the case where each $f_i$ is the identity function (i.e., $f$ is a linear model). In this case, determining the minimum and maximum values in each domain is straightforward and can be done in polynomial time. For cubic splines, this result extends naturally, as it involves iterating over each defined region of the piecewise polynomial $f_i$. For each region, we evaluate the value of $f_i$ at the domain edges and at the extreme points, which can be found by solving a polynomial equation of degree 3 (achievable in polynomial time). Among these values (edges and polynomial extremes), the minimum and maximum are selected. If $\mathcal{X}_i$ is a general discrete set, we iterate over the continuous region representing $\mathcal{X}_i$, find the extreme points as before, and for each extreme point, identify the two nearest discrete points within $\mathcal{X}_i$ — one above and one below. From this set of viable extreme points of the discrete domain $\mathcal{X}_i$, we then select the minimum and maximum values.

We note that this proof applies not only to Smooth GAMs but to any model where the minimum and maximum attainable values of each $f_i$ can be computed in polynomial time. This includes, for example, decision trees. In the case where $f_i$ is a decision tree, we iterate over all leaf edges (representing all viable values of $f_i$) and select the minimum and maximum values from these leaves. This process can also be completed in polynomial time with respect to the size of the tree.

□

# F   Proof of Proposition 2

**Proposition 2.** *Given a Smooth GAM $f$ over a continuous input space, then the FR query can be obtained in polynomial time.*

*Proof.* We will prove this for a more general class that includes any GAM $f$ defined over continuous inputs, where the output $f(\mathbf{x})$ is continuous, and for which both $\min\{\beta_i f(i)\}$ and $\max\{\beta_i f(i)\}$ can be computed in polynomial time for any feature $i$. The polynomial-time computation of $\min\{\beta_i f(i)\}$ and $\max\{\beta_i f(i)\}$ for splines has been proven in Lemma 1. Furthermore, since splines are defined as piecewise polynomials, it is clear that $f(\mathbf{x})$ is a continuous function. Therefore, proving the complexity results for the FR query for this family of classifiers will also encompass GAMs defined over splines.

We first note that by definition a feature $i$ is redundant with respect to $f$ if and only if for any $\mathbf{x}', \mathbf{x}_i, \mathbf{z}_i$ it holds that $f(\mathbf{x}'_{[n] \setminus \{i\}}; \mathbf{x}_i) = f(\mathbf{x}'_{[n] \setminus \{i\}}; \mathbf{z}_i)$ (or in other words, changing the value of feature $i$ cannot cause the classification to change. We will prove the following claim:

**Lemma 7.** *Let there be some GAM $f$ where both the input domain $\mathcal{X}$ is continuous, and the output $f(\mathbf{x})$ is continuous. Let us assume that $i$ is redundant with respect to $\langle f \rangle$ and that $f$ is not trivial (in*

*other words, a function that always outputs 1 or always outputs 0). Then, it holds that for all $\boldsymbol{x}_i \in \mathcal{X}_i$, $\beta_i f_i(\boldsymbol{x}_i) = 0$.*

*Proof.* First we recall that by definition $i$ is redundant with respect to $f$ iff for any $\mathbf{x}', \mathbf{x}_i, \mathbf{z}_i$ it holds that $f(\mathbf{x}'_{[n]\setminus\{i\}}; \mathbf{x}_i) = f(\mathbf{x}'_{[n]\setminus\{i\}}; \mathbf{x}_i)$. In other words, this implies that:

$$
\begin{aligned}
&[[\beta_0 + \beta_1 f_1(\mathbf{x}_1) + \ldots + \beta_{i-1} f_{i-1}(\mathbf{x}_{i-1}) + \beta_{i+1} f_{i+1}(\mathbf{x}_{i+1}) + \ldots + \beta_k f_k(\mathbf{x}_k) \geq 0] \wedge \\
&\hspace{6cm} {}^{\backprime}[\beta_0 + \beta_1 f_1(\mathbf{x}_1) + \ldots + f_k(\mathbf{x}_k) \geq 0]] \\
&\vee [[\beta_0 + \beta_1 f_1(\mathbf{x}_1) + \ldots + \beta_{i-1} f_{i-1}(\mathbf{x}_{i-1}) + \beta_{i+1} f_{i+1}(\mathbf{x}_{i+1}) + \ldots + \beta_k f_k(\mathbf{x}_k) < 0] \wedge \\
&\hspace{6cm} {}^{\backprime}[\beta_0 + \beta_1 f_1(\mathbf{x}_1) + \ldots + f_k(\mathbf{x}_k) < 0]]
\end{aligned}
\tag{13}
$$

Let us assume for contradiction that for any $\mathbf{x}$, the term $\beta_0 + \beta_1 f_1(\mathbf{x}_1) + \ldots + \beta_{i-1} f_{i-1}(\mathbf{x}_{i-1}) + \beta_{i+1} f_{i+1}(\mathbf{x}_{i+1}) + \ldots + \beta_k f_k(\mathbf{x}_k)$ is always strictly positive or always strictly negative. Since $i$ is redundant, it follows that adding the term $\beta_i f_i(\mathbf{x}_i)$ to the rest of the sum cannot cause the classification to "flip," meaning the overall term cannot change from positive to negative or vice versa. This implies that the entire sum $\beta_0 + \beta_1 f_1(\mathbf{x}_1) + \ldots + \beta_k f_k(\mathbf{x}_k)$ for any $\mathbf{x}$ is always positive or always negative, which contradicts the assumption that $f$ is non-trivial. Therefore, there must necessarily exist some $\mathbf{x}, \mathbf{z} \in \mathcal{X}$ such that $\beta_0 + \beta_1 f_1(\mathbf{x}_1) + \ldots + \beta_{i-1} f_{i-1}(\mathbf{x}_{i-1}) + \beta_{i+1} f_{i+1}(\mathbf{x}_{i+1}) + \ldots + \beta_k f_k(\mathbf{x}_k)$ is positive, and $\beta_0 + \beta_1 f_1(\mathbf{z}_1) + \ldots + \beta_{i-1} f_{i-1}(\mathbf{z}_{i-1}) + \beta_{i+1} f_{i+1}(\mathbf{z}_{i+1}) + \ldots + \beta_k f_k(\mathbf{z}_k)$ is negative (or vice versa).

Moreover, since $\mathcal{X}$ is continuous, we can select $\mathbf{x}$ and $\mathbf{z}$ such that $\beta_0 + \beta_1 f_1(\mathbf{x}_1) + \ldots + \beta_{i-1} f_{i-1}(\mathbf{x}_{i-1}) + \beta_{i+1} f_{i+1}(\mathbf{x}_{i+1}) + \ldots + \beta_k f_k(\mathbf{x}_k)$ and $\beta_0 + \beta_1 f_1(\mathbf{z}_1) + \ldots + \beta_{i-1} f_{i-1}(\mathbf{z}_{i-1}) + \beta_{i+1} f_{i+1}(\mathbf{z}_{i+1}) + \ldots + \beta_k f_k(\mathbf{z}_k)$ are infinitesimally close to the decision boundary. We will denote these points as either $0^+$ or $0^-$, representing an infinitesimally close value that is either above or below 0. Now, assuming there exists some $\mathbf{x}'_i$ such that $\beta_i f_i(\mathbf{x}'_i)$ is negative, we can take an infinitesimally small value for $\beta_0 + \beta_1 f_1(\mathbf{x}_1) + \ldots + \beta i - 1 f_{i-1}(\mathbf{x}_{i-1}) + \beta_{i+1} f_{i+1}(\mathbf{x}_{i+1}) + \ldots + \beta_k f_k(\mathbf{x}_k)$ above the decision boundary $(0^+)$ such that adding $\beta_i f_i(\mathbf{x}'_i)$ causes the decision to become negative. The analogous condition holds for the opposite direction of the decision boundary (assuming there exists some $\mathbf{x}'_i$ for which $\beta_i f_i(\mathbf{x}'_i)$ is strictly negative). Consequently, the only way for $i$ to be redundant is if, for every $\mathbf{x}'_i \in \mathcal{X}_i$, it holds that $\beta_i f_i(\mathbf{x}'_i) = 0$, thereby concluding the proof of the lemma.

$\square$

We will now use the previous Lemma to describe a simple polynomial-time algorithm for solving this task. The algorithm iterates over all features $i$ from 1 to $n$. Since $f_i$ is a spline, we can compute the minimal and maximal values $\min \beta_i f_i(\mathbf{x}_i)$ and $\max \beta_i f_i(\mathbf{x}_i)$ in polynomial time. If either both terms are equal to 0 or $\beta_i = 0$, we return that $i$ is redundant. Otherwise, we return that it is not. The correctness follows from the correctness of Lemma 7, the continuity of $\mathcal{X}$, and the continuity of $f(\mathbf{x})$ (a result of working with splines). This concludes the proof.

$\square$

# G   Proof of Proposition 2

**Proposition 3.** *Given any regression GAM with enumerable discrete inputs, or given a regression Smooth GAM with discrete or continuous inputs, then SHAP query is solvable in polynomial time.*

*Proof.* We divide the proof into three separate lemmas. First, we establish the claim for general GAMs under the enumerable discrete setting. Next, we address Smooth GAMs over general discrete domains. Finally, we prove the claim for Smooth GAMs over continuous domains.

**Lemma 8.** *Assuming $f$ is a regression GAM over an enumerable discrete input space $\mathbb{F}$, then solving SHAP for $\langle f, \boldsymbol{x}, i \rangle$ can be solved in polynomial time.*

*Proof.* For a certain value function $v(S)$, given the assumption of feature independence and the additive structure of $f$, we have:

$$v(S) = \mathbb{E}_{\mathbf{z} \sim \mathcal{D}_p}[f(\mathbf{z})|\mathbf{z}_S = \mathbf{x}_S] =$$
$$\mathbb{E}_{\mathbf{z}_{\bar{S}}}[f(\mathbf{x}_S; \mathbf{z}_{\bar{S}})] =$$
$$\mathbb{E}_{\mathbf{z} \sim \mathcal{D}_p}[f(\mathbf{x}_S; \mathbf{z}_{\bar{S}})] =$$
$$\sum_{\mathbf{z} \in \mathbb{F}} \Big( \prod_{\ell \in [n], \mathbf{z}_\ell = j} p(\ell, j) \Big) \cdot f(\mathbf{x}_S; \mathbf{z}_{\bar{S}}) =$$
$$\sum_{\mathbf{z} \in \mathbb{F}} \Big( \prod_{\ell \in [n], \mathbf{z}_\ell = j} p(\ell, j) \Big) \cdot \Big( \sum_{\ell \in S} \beta_\ell f_\ell(\mathbf{x}_\ell) + \sum_{j \in \bar{S}} \beta_\ell f_j(\mathbf{z}_j) + \beta_0 \Big) =$$
$$\sum_{\ell \in S} \beta_\ell f_\ell(\mathbf{x}_\ell) + \sum_{j \in \bar{S}} \mathbb{E}_{\mathbf{z} \sim \mathcal{D}_p}[\beta_j f_j(\mathbf{z}_j)] + \beta_0 \tag{14}$$

Now, we have that:

$$v(S \cup \{i\}) - v(S) =$$
$$\Big( \sum_{\ell \in S \cup \{l\}} \beta_\ell f_\ell(\mathbf{x}_\ell) + \sum_{j \in \bar{S} \setminus \{i\}} \mathbb{E}_{\mathbf{z} \sim \mathcal{D}_p}[\beta_j f_j(\mathbf{z}_j)] + \beta_0 \Big) -$$
$$\Big( \sum_{\ell \in S} \beta_\ell f_\ell(\mathbf{x}_\ell) + \sum_{j \in \bar{S}} \mathbb{E}_{\mathbf{z} \sim \mathcal{D}_p}[\beta_j f_j(\mathbf{z}_j)] + \beta_0 \Big) = \tag{15}$$
$$\beta_i f_i(\mathbf{x}_i) - \mathbb{E}_{\mathbf{z} \sim \mathcal{D}_p}[\beta_i f_i(\mathbf{x}_i)] =$$
$$\beta_i (f_i(\mathbf{x}_i) - \mathbb{E}_{\mathbf{z} \sim \mathcal{D}_p}(f_i(\mathbf{z}_i)))$$

By incorporating it into the SHAP formulation, we obtain:

$$\phi_i(f, \mathbf{x}) = \sum_{S \subseteq [n] \setminus \{i\}} \frac{|S|!(n - |S| - 1)!}{n!} (v(S \cup \{i\}) - v(S)) =$$
$$\sum_{S \subseteq [n] \setminus \{i\}} \frac{|S|!(n - |S| - 1)!}{n!} (\beta_i f_i(\mathbf{x}_i) - \mathbb{E}_{\mathbf{z} \sim \mathcal{D}_p}[\beta_i f_i(\mathbf{x}_i)]) = \tag{16}$$
$$\frac{\beta_i}{n} (f_i(\mathbf{x}_i) - \mathbb{E}_{\mathbf{z} \sim \mathcal{D}_p}(f_i(\mathbf{z}_i)))$$

where the last equation follows from the fact that the inner product does not depend on $S$, and from the well-known result that the sum of the Shapley values, $\sum_{S \subseteq [n] \setminus \{i\}} \frac{|S|!(n - |S| - 1)!}{n!}$, simplifies to $\frac{1}{n}$ through a telescoping sum. By definition, $f_i(\mathbf{x}_i)$ can be computed in polynomial time. Furthermore, since the domain $\mathcal{X}_i$ is enumerable discrete, $\mathbb{E}_{\mathbf{z} \sim \mathcal{D}_p}[f_i(\mathbf{z}_i)]$ can also be computed in polynomial time by evaluating $\sum_{j \in \mathcal{X}_i} p(i, j) f_i(\mathbf{z}_i)$ (where the equivalence holds again by utilizing the feature independence property), thereby completing the proof.

$\square$

**Lemma 9.** *Given a GAM $f$ over a general discrete input space, obtaining the SHAP query for regression can be solved in polynomial time.*

*Proof.* We note that until our last step — proving tractability for the enumerable discrete setting in Lemma 8, which involved showing that $\mathbb{E}_{\mathbf{z} \sim \mathcal{D}_p}[f_i(\mathbf{z}_i)]$ can be computed in polynomial time (by evaluating $\sum_{j \in \mathcal{X}_i} p(i, j) f_i(\mathbf{z}_i)$) — we did not assume explicit enumerability. Thus, all results up to this point also apply to the general discrete case. In the general setting, correctness follows directly from the poly-summability property of the distribution $\mathcal{D}_p$, thereby establishing tractability.

$\square$

**Lemma 10.** *Given a GAM $f$ over a continuous input space with poly-integrable components, obtaining the SHAP for regression can be solved in polynomial time.*

*Proof.* The proof will be similar to the discrete scenario. For a certain value function $v(S)$, given the assumption of feature independence and the additive structure of $f$, we have:

$$v(S) = \mathbb{E}_{\mathbf{z} \sim \mathcal{D}_p}[f(\mathbf{z}) | \mathbf{z}_S = \mathbf{x}_S] =$$
$$\mathbb{E}_{\mathbf{z}_{\bar{S}}}[f(\mathbf{x}_S; \mathbf{z}_{\bar{S}})] =$$
$$\mathbb{E}_{\mathbf{z} \sim \mathcal{D}_p}[f(\mathbf{x}_S; \mathbf{z}_{\bar{S}})] =$$
$$\int_{\mathbf{z} \in \mathcal{X}} p(\mathbf{z}) \cdot f(\mathbf{x}_S; \mathbf{z}_{\bar{S}}) =$$
$$\int_{\mathbf{z} \in \mathcal{X}} \left( p(\mathbf{z}) \cdot \left( \sum_{\ell \in S} \beta_\ell f_\ell(\mathbf{x}_\ell) + \sum_{j \in \bar{S}} \beta_\ell f_j(\mathbf{z}_j) + \beta_0 \right) \right) =$$
$$\sum_{\ell \in S} \beta_\ell f_\ell(\mathbf{x}_\ell) + \sum_{j \in \bar{S}} \mathbb{E}_{\mathbf{z} \sim \mathcal{D}_p}[\beta_j f_j(\mathbf{z}_j)] + \beta_0 \qquad (17)$$

Now, the following condition holds:

$$v(S \cup \{i\}) - v(S) =$$
$$\left( \sum_{\ell \in S \cup \{l\}} \beta_\ell f_\ell(\mathbf{x}_\ell) + \sum_{j \in \bar{S} \setminus \{i\}} \mathbb{E}_{\mathbf{z} \sim \mathcal{D}_p}[\beta_j f_j(\mathbf{z}_j)] + \beta_0 \right) -$$
$$\left( \sum_{\ell \in S} \beta_\ell f_\ell(\mathbf{x}_\ell) + \sum_{j \in \bar{S}} \mathbb{E}_{\mathbf{z} \sim \mathcal{D}_p}[\beta_j f_j(\mathbf{z}_j)] + \beta_0 \right) = \qquad (18)$$
$$\beta_i f_i(\mathbf{x}_i) - \mathbb{E}_{\mathbf{z} \sim \mathcal{D}_p}[\beta_i f_i(\mathbf{x}_i)] =$$
$$\beta_i (f_i(\mathbf{x}_i) - \mathbb{E}_{\mathbf{z} \sim \mathcal{D}_p}(f_i(\mathbf{z}_i)))$$

Incorporating it into the SHAP formulation yields:

$$\phi_i(f, \mathbf{x}) = \sum_{S \subseteq [n] \setminus \{i\}} \frac{|S|!(n - |S| - 1)!}{n!} (v(S \cup \{i\}) - v(S)) =$$
$$\sum_{S \subseteq [n] \setminus \{i\}} \frac{|S|!(n - |S| - 1)!}{n!} (\beta_i f_i(\mathbf{x}_i) - \mathbb{E}_{\mathbf{z} \sim \mathcal{D}_p}[\beta_i f_i(\mathbf{x}_i)]) = \qquad (19)$$
$$\frac{\beta_i}{n} (f_i(\mathbf{x}_i) - \mathbb{E}_{\mathbf{z} \sim \mathcal{D}_p}(f_i(\mathbf{z}_i)))$$

The final equality follows from the fact that the inner product is independent of $S$, along with the well-known identity that the sum of Shapley coefficients, $\sum_{S \subseteq [n] \setminus i} \frac{|S|!(n - |S| - 1)!}{n!}$, evaluates to $\frac{1}{n}$ via a telescoping argument. By definition, $f_i(\mathbf{x}_i)$ is computable in polynomial time. Thus, the question of whether the SHAP value is polynomial-time computable reduces to whether $\mathbb{E}_{\mathbf{z} \sim \mathcal{D}_p}[f_i(\mathbf{z}_i)]$ can be computed in polynomial time. Under the feature independence assumption, we observe that:

$$\mathbb{E}_{\mathbf{z} \sim \mathcal{D}_p}[f_i(\mathbf{z}_i)] := \int_{\mathbf{z}_i \in \mathcal{X}_i} p(\mathbf{z}) f_i(\mathbf{z}_i) = \int_{\mathbf{z}_i \in \mathcal{X}_i} p_i(\mathbf{z}_i) f_i(\mathbf{z}_i) \qquad (20)$$

Since $f_i(\mathbf{z}_i)$ is a smooth function bounded by degree 3, and $p(\mathbf{z}_i)$ is poly-integrable, it follows by definition that $\mathbb{E}_{\mathbf{z} \sim \mathcal{D}_p}[f_i(\mathbf{z}_i)]$ can be computed in polynomial time by evaluating the integral $\int_{\mathbf{z}_i \in \mathcal{X}_i} p_i(\mathbf{z}_i) f_i(\mathbf{z}_i)$. This completes the proof.

$\square$

# H   Proof of Proposition 4

**Proposition 4.** *Given any unary-encoded GAM over a discrete, enumerable input domain with polynomially bounded output, the CC and SHAP-C queries admit a pseudo-polynomial-time algorithm.*

We will divide the proof into two lemmas, addressing the CC and SHAP queries separately, beginning with the CC case.

**Lemma 11.** *Given any unary-encoded GAM over a discrete, enumerable input domain with polynomially bounded output, the CC query admits a pseudo-polynomial-time algorithm.*

*Proof.* We prove this by reducing the CC problem when weights are provided in unary to a variant of the counting version of the classic *#Knapsack* problem, which we refer to as *#Multi-Choice Knapsack*. We then show that *#Multi-Choice Knapsack* can be solved via a dynamic programming algorithm. Our proof follows a similar structure to that of [16], who addressed linear models over binary domains, but introduces additional complexity to handle arbitrary enumerable discrete input configurations. In particular, whereas the proof in [16] addresses this problem via a reduction to *#Knapsack*, our approach reduces it to the *#Multi-Choice Knapsack* setting.

---

#**Multi-Choice Knapsack**:
**Input**: $w := (w_1, \ldots, w_n)$ where each $w_i$ is a non-negative integer weight, a vector of sets $(S_1, \ldots, S_n)$ where each set $S_i$ contains a fixed number of associated integers, and a target integer $C$ which denotes the capacity.
**Output**: $|\{(x_1, \ldots, x_n) \in S_1 \times \ldots \times S_n \mid \sum_{i \leq n} w_i x_i \leq C\}|$

---

We observe that this problem resembles the classic *#Knapsack* problem — the counting variant of Knapsack — except that each variable $x_i$ ranges over a set $S_i$ of enumerable integers, rather than being restricted to $\{0, 1\}$. This range of possible values corresponds to the outputs of the GAM. Since we assume that the GAM uses an enumerably discrete unary input encoding and that its outputs are polynomially bounded, it follows that the set of possible outputs is itself polynomially bounded with respect to the input encoding (and therefore also unary). The work of [16] reduces the CC query for Perceptrons (i.e., linear models over binary inputs with unary weights) to the classic *#Knapsack* problem via a polynomial-time reduction, where $C$ is set to the bias $b$ of the linear model. We observe that applying the same reduction — except replacing binary inputs with sets of enumerable discrete values representing the enumerable set of unary-encoded possible outputs of the GAM — yields an instance of the *#Multi-Choice Knapsack* problem instead (and the target integer $C$ equals the intercept $\beta_0$ of the GAM instead of the bias $b$ of the linear model). Thus, it remains to show that *#Multi-Choice Knapsack* is solvable in pseudo-polynomial time, which we do by adapting the standard dynamic programming algorithm typically used for this problem (and also discussed in [16]).

The algorithm will be constructed as follows: given an instance of *#Multi-Choice Knapsack* which includes $w := (w_1, \ldots, w_n)$, $(S_1, \ldots, S_n)$, and $C$, we can define the following quantity:

$$DP[i][C] := |\{(x_1, \ldots, x_n) \in S_1 \times \ldots \times S_n \mid \sum_{i \leq n} w_i x_i \leq C\}| \tag{21}$$

The final result for the *#Multi-Choice Knapsack* (and consequently for the *CC* query) is given by the value of $DP[n][b]$. This can be computed using an iterative dynamic programming approach based on the following inductive step:

$$DP[i+1][C] = DP[i][C] + \sum_{s \in S_{i+1}} DP[i][C-s] \tag{22}$$

We carry out this iterative computation starting from the standard dynamic programming base case (as in the classic Knapsack and #Knapsack algorithms, and also noted in [16]): $DP[0][\alpha] = 0$ for all $\alpha < 0$ and $DP[0][\alpha] = 1$ for all $\alpha \geq 0$. Given the pseudo-polynomial nature of the problem, a polynomial number of iterations suffices to compute the final value $DP[n][b]$, which yields the solution to the *#Multi-Choice Knapsack* and, consequently, the answer to the *CC* query in this setting.

$\square$

**Lemma 12.** *Given any unary-encoded GAM over a discrete, enumerable input domain with polynomially bounded output, the SHAP query admits a pseudo-polynomial-time algorithm.*

*Proof.* We build on the proof introduced by [102] which showed that the computation of SHAP for a model $f$ and some input $\mathbf{x}$ can be reduced to the computation of $\mathbb{E}_{\mathbf{z} \sim \mathcal{D}_p}[f(\mathbf{z})]$ in polynomial time, given that we assume feature independence. Hence, we are only left to prove that $\mathbb{E}_{\mathbf{z} \sim \mathcal{D}_p}[f(\mathbf{z})]$ can

be obtained in pseudo-polynomial time for GAMs over enumerable discrete input spaces. We note that the following holds:

$$\mathbb{E}_{\mathbf{z} \sim \mathcal{D}_p}[f(\mathbf{z})] = \mathbb{E}_{\mathbf{z} \sim \mathcal{D}_p}[\text{step}(\beta_0 + \beta_1 \cdot f_1(\mathbf{z}_1) + \dots \beta_n \cdot f_n(\mathbf{z}_n))] \tag{23}$$

Since we assume that the input space $\mathcal{X}$ is enumerable discrete, computing $\mathbb{E}_{\mathbf{z} \sim \mathcal{D}p}[f(\mathbf{z})]$ reduces to counting the number of assignments $(x_1, \dots, x_n) \in S_1 \times \dots \times S_n$ such that $\beta_0 + \beta_1 \cdot f_1(\mathbf{z}_1) + \dots + \beta_n \cdot f_n(\mathbf{z}_n) \geq 0$. This is equivalent to counting assignments where $\sum i = 1^n \beta_i \cdot f_i(\mathbf{z}_i) \geq -\beta_0$. Setting the threshold $C := \beta_0$, the problem becomes an instance of the *#Multi-Choice Knapsack* problem described in Lemma H. As shown in that lemma, this problem can be solved in pseudo-polynomial time via a dynamic programming approach. Therefore, the SHAP query for classification over GAMs with enumerable discrete input domains is also solvable in pseudo-polynomial time.

$\square$

# I   Proof of Proposition 5

**Proposition 5.** *Let there be a NAM or an EBM over a discrete or contentious input space, then solving the CSR, and MSR queries are coNP-Complete, and the MCR query is NP-Complete.*

*Proof.* We will divide the proof into several lemmas, beginning with NAMs and then proceeding to EBMs. For each model, we will first establish the results for CSR, followed by MCR, and finally the MSR query. We will also start with continuous domains and then explain how to extend the results to general discrete domains in each case.

**Lemma 13.** *Given a NAM $f$, an input $\boldsymbol{x} \in \mathbb{F}$, and an integer $k \in \mathbb{N}$, where $f$ is defined over either a continuous input space or a general discrete input space, then obtaining the CSR query is coNP-Complete.*

*Proof.* **Membership.** In the general discrete setting, the proof is straightforward. One can guess a set of $|\overline{S}|$ assignments corresponding to the features in $\overline{S}$. This is feasible because the domain of each feature $i$, denoted $\mathcal{X}_i$, is defined by a minimum and maximum value, each represented with at most $q$ bits. Moreover, any value in the domain can also be expressed with at most $q$ bits. Therefore, any value for a feature $i$ in $\overline{S}$ can be determined by guessing the bit values for each feature representation. Denote the guessed values for $\overline{S}$ as $\mathbf{z}_{\bar{S}}$, and select arbitrary assignments for the features in $S$ (chosen from their respective domains).

This allows us to construct a vector $\mathbf{z}$ that includes both the guessed values for $\overline{S}$ and the arbitrarily chosen values for $S$. We then verify whether $f(\mathbf{x}_S; \mathbf{z}_{\bar{S}}) \neq f(\mathbf{x})$, which can be done in polynomial time. If this condition holds, it implies that $S$ is not a sufficient reason for $\langle f, \mathbf{x} \rangle$, thereby establishing membership in coNP.

In the continuous setting, the situation becomes more intricate, as guessing a witness assignment for the input is not feasible. This is because the size of the encoding is no longer polynomial and may not even be finite. However, as shown in a similar proof from [96], for a neural network with ReLU activations, one can, instead of guessing the input assignment, guess the activation status of each neuron (either active or inactive).

Once the activation status of each neuron is established, and assuming all remaining constraints are linear, finding a satisfying assignment for the neural network reduces to solving a linear programming problem. In our case, this indeed holds, since it involves setting lower and upper bounds for each input (corresponding to the minimum and maximum permissible values in $\mathcal{X}_i$ for each feature $i \in \overline{S}$) and equality constraints for features in $S$. For the output constraint, we impose a condition that it is either greater or less than zero, depending on the opposite of the classification result for $f(\mathbf{x})$. If this linear program yields a feasible solution, it demonstrates that $f(\mathbf{x}_S; \mathbf{z}_{\bar{S}}) \neq f(\mathbf{x})$, hence proving membership in coNP.

**Hardness.** We begin by establishing hardness through the (complementary of) the neural network verification problem, which has been examined in [69] and [96], among numerous other studies. This problem, known to be NP-Complete [69, 96], is formalized as follows:

> **Neural Network Verification**:
> **Input**: A neural network $f : \mathbb{R}^d \to \mathbb{R}^c$, a bounded continuous domain $\mathcal{X}$ over the input space, and a bounded continuous domain $\mathcal{Y}$ over the output space.
> **Output**: *Yes*, if there exists some $\mathbf{x} \in \mathcal{X}$ and some $\mathbf{y} \in \mathcal{Y}$ such that $f(x) = y$, and *No* otherwise

We will prove hardness by reducing from the (complementry of) the Neural Network verification problem, which is known to be NP-Complete [69, 96]. We note that while the authors in both [69] and [96] describe this as a more general problem for wny piec-wise linear specification over the input and output (and not only for bounded domains), in practice the hardness result that were proven from the classic *SAT* problem were performed for specificacations of this form (bounding both the input and the outputs). Hence, the NP-Hardness of these problems holds from these reductions. Moreover, as noted by [96] — hardness remains for neural networks consisting of a *single* output neuron, and hence we can assume that our neural network is actually of the form $f : \mathbb{R}^d \to \mathbb{R}$.

We will begin by establishing an intermediate proof for our claim, which is even a stronger one that was obtained by [96]. Particularly, we will prove that the neural network verification problem remains NP-Hard even for a model with both a single input and a single output. We will later demonstrate why this hardness result also applies to the general discrete input setting.

**Lemma 14.** *Given a neural network $f : \mathbb{R} \to \mathbb{R}$, a bounded continuous input domain $\mathcal{X}$, and a bounded continuous output domain $\mathcal{Y}$, then solving the neural network verification problem over $f$ is NP-Hard.*

We will begin by establishing hardness for the continuous domain and then demonstrate why these results extend to the general discrete domain. Now, let there be a neural network $f : \mathbb{R}^d \to \mathbb{R}$ with a bounded continuous input domain $\mathcal{X}$, and a bounded continuous output domain $\mathcal{Y}$. First, we will show how to reduce $f$ into a model $f' : \mathbb{R} \to \mathbb{R}$, with some continuous bounded domain $\mathcal{X}'$ over the (individual) input of $f'$ and the same output domain $\mathcal{Y}$, for which it holds that there exists some $\mathbf{x} \in \mathcal{X}, \mathbf{y} \in \mathcal{Y}$ for which $f(\mathbf{x}) = \mathbf{y}$ if and only if there also exists some $\mathbf{x}' \in \mathcal{X}'$ for which $f'(\mathbf{x}') = \mathbf{y}$.

We first note that given some value $o_i$ that is obtained by some arbitraty neuron in an MLP, it is possible to construct the calculation of $o_j := |o_i|$ in the preceding layers by incorporating the following procedure:

$$o_j = \text{ReLU}(o_i) + (-1)\text{ReLU}(-o_i) = |o_i| \tag{24}$$

which can be computed by simply adding two additional neurons in the preceding layers with $1$ and $-1$ weight and connecting them both to another preceding layer. We hence, will consider, for simplicity that for any $o_i$, $o_j := |o_i|$ can be computed by a constant number of preceding layers. We also note that given some $o_i$ we can simply compute:

$$o_j = \text{ReLU}(o_i) + \text{ReLU}(-o_i) = o_i \tag{25}$$

meaning that a single value of a neuron can be passed to the subsequent layers with a simple identity transformation with that construction. We will hence regard the identity construction as a valid one as well. Lastly we regard the following construction:

$$o_j = \text{ReLU}(o_i) - 1 = \max(o_i, 0) \tag{26}$$

We observe that, as previously mentioned, the proof in [96] was derived from the *CNF-SAT* problem, initially assuming the input space $\mathcal{X} := \{0, 1\}^n$. Consequently, we will specifically demonstrate hardness for the neural network verification problem involving a model of the form $f' : \mathbb{R} \to \mathbb{R}$ by reducing it from the neural network verification problem over a model of the form $f : \{0, 1\}^n \to \mathbb{R}$.

Furthermore, we observe that the input domain required for our model $f$ is a bounded continuous domain $\mathcal{X}$, defined by minimum and maximum values represented in binary. For simplicity and clarity, we demonstrate how to reduce a model $f$, defined over the domain $\mathcal{X} := [0, 2^n - 1]$, to the domain $\{0, 1\}^n$. This reduction is achieved through a binary search construction over $[0, 2^{n-1}]$ and can be generalized to any bounded continuous domain $\mathcal{X}$, where the minimum value of $\mathcal{X}$ corresponds to 0 and the maximum value corresponds to $2^{n-1}$.

**Reducing the continuous domain** $[0, 2^n - 1]$ **for an individual feature to the discrete domain** $\{0, 1\}^n$ **for** $n$ **features.** The construction of $f'$ is as follows. Following the first input neuron $o^0$ we will connect it to a neuron in the preceding layers which calculates: $o_0^1 := \text{ReLU}(o^0 - 2^{n-1})$ and $o_1^1 := \text{ReLU}(o_0^1) - \text{ReLU}(o_0^1 - 2^{n-1})$. We connect $o_1^1$ to the first neuron from layer $o_1'$ via a linear transformation, i.e., $o_1' := o_1^1$. We also set $o_2^1 := o_0^1 - \text{ReLU}(o_1^1)$.

We now move on and construct another hidden layer over which we compute $o_0^2 := \text{ReLU}(o_1^1 - 2^{n-2})$ and $o_1^2 := \text{ReLU}(o_1^1) - \text{ReLU}(o_1^1 - 2^{n-2})$. We connect $o_1^2$ to the second neuron in the $o'$ layer via a linear transformation, i.e., we compute $o_2' := o_1^2$. We then again set $o_2^2 := o_0^2 - \text{ReLU}(o_1^2)$ and $o_0^3 := \text{ReLU}(o_1^2 - 2^{n-3})$, and continue performing this entire process recursively, i.e., for the $i$'th iteration in this process we construct $o_0^i := \text{ReLU}(o_2^{i-1} - 2^{n-i})$, $o_1^i := \text{ReLU}(o_1^i) - \text{ReLU}(o_0^i - 2^{n-i})$, $o_2^i := o_0^i - \text{ReLU}(o_1^i)$ and connect input $i$ in the $o'$ layer to $o_1^i$ via a linear transformation, i.e.,: $o'^i := o_1^i$.

The following construction takes some arbitrary assignment $o_0$ in the range $[0, 2^n]$ and performs a set of binary decisions, each time decreasing $2^{i-1}$ and continuing with the remainder. This process hence gives us in the first neuron $o_1'$ a positive value iff $o_0$ is larger than $2^{n-1}$, and otherwise will be assigned a 0. The second neuron $o_2'$ will get a positive value iff the remainder after decreasing $2^{n-1}$ is larger than 0, and otherwise it will be set to 0. The third neuron $o_3'$ will get a positive value iff the remainder of what is left, while decreasing $2^{n-2}$ from it is larger than 0, and otherwise, it will be assigned 0. This continues for every $i$. It is straightforward to show that for a value $o' = \lfloor x \rfloor$ for some $0 \geq x \geq 2^n$ it holds that we can equivalentley describe $o'$ as a binary vector of size $n$ which is equivalent to the binary value of $x$, where each "1" value in place $i$ in the binary vector corresponds to some positive assignment in place $i$ in the vector $o'$, and each "0" value in place $i$ in the binary vector corresponds to some 0 assignment in the $o'$ vector.

We now will describe an additional transformation over $o'$ that we do befor connecting it to get $o''$ which we will connect to the first hidden layer of the original one in $f$ (and hence to the preceding layers in $f$ as well). First, let us observe that for the following construction:

$$z := \text{ReLU}(1 - x) + \text{ReLU}(x - 1) \tag{27}$$

it holds that if $0 \leq x \leq 1$ then $z = 1$. We hence can define the additional hidden layer $o''$ which comes after $o'$ such that for each $o_i'$ we define:

$$o_i'' := \text{ReLU}(1 - \frac{1}{2^n} o_i') + \text{ReLU}(\frac{1}{2^n} o_i' - 1) \tag{28}$$

Since the range of the original individual input of $f'$: $o^0$ is in $[0, 2^n]$, then we have that each remainder that is propagated into the $o'$ layer is also in the range $[0, 2^n]$. We hence get that for any positive instance that was propagated into $o'$ (and represents its corresponding binary vector) its output will be equal to exactly 1 in $o''$.

We connect layer $o''$ to the remaining hidden layers from the original $f$. We note that the continuous domain $[0, 2^n]$ over the single input neuron in $o^0$ in $f$ is mapped exactly to any possible binary vector $\{0, 1\}^n$ in layer $o''$. The preceding layers after $o''$ are identical between $f$ and $f'$ and hence we have that:

$$\max_{x \in [0, 2^n - 1]} f'(x) = \max_{x \in \{0,1\}^n} f(x) \quad and \quad \min_{x \in [0, 2^n - 1]} f'(x) = \min_{x \in \{0,1\}^n} f(x) \tag{29}$$

We connect layer $o''$ to the remaining hidden layers from the original $f$. We note that the continuous domain $[0, 2^n]$ over the single input neuron in $o^0$ in $f$ is mapped exactly to any possible binary vector $\{0, 1\}^n$ in layer $o''$. The preceding layers after $o''$ are identical between $f$ and $f'$. Thus, for any $\mathbf{x} \in [0, 2^{n-1}]$, we can consider the binary representation of the lower value within the range that maps to $\{0, 1\}^n$ (denoted as $\mathbf{x}'$), satisfying $f'(\mathbf{x}') = f(\mathbf{x})$. This demonstrates more specifically that there exists $\mathbf{x} \in \mathcal{X}, \mathbf{y} \in \mathcal{Y}$ such that $f(\mathbf{x}) = \mathbf{y}$ if and only if there exists some $\mathbf{x}' \in \mathcal{X}' = \{0, 1\}^n$ for which $f(\mathbf{x}) = f'(\mathbf{x}') = \mathbf{y}$. This hence concludes the proof of our Lemma.

$\square$

**Extending the proof to the general discrete setting.** We observe that the construction we developed for $f'$ relies on performing a binary search over the features within $[0, 2^{n-1}]$, mapping them into

the discrete input space $0, 1^n$ by associating each value with its corresponding lower value in the binary domain. If we consider the domain $\mathcal{X}' := [0, 2^{n-1}]$ to consist solely of discrete binary values represented with $n$ bits, the mapping functions identically. In this case, instead of assigning a lower value for each mapping, the values are mapped exactly. Consequently, it is straightforward to conclude that the hardness proof remains valid when $\mathcal{X}$ is assumed to be general discrete.

**Finalizing the proof.** We have established that the neural network verification problem for a neural network with one input and one output, where the input domain is either continuous or general discrete, is NP-Hard. We now aim to leverage this hardness result to demonstrate that the CSR query for a NAM defined over either a general discrete or a continuous domain is coNP-Hard. We observe that the hardness proof we derived applies to a specific scenario in which the output specification $\mathcal{Y}$ mandates that the prediction of $f(\mathbf{x})$ must be positive (i.e., $f(\mathbf{x}) \geq 0$). Consequently, we will demonstrate hardness by reducing such an instance — a neural network with a single input and output, a bounded input domain $\mathcal{X}$, where the task is to determine whether $f(\mathbf{x}) \geq 0$ — to the (complement of) the CSR query.

Given a model $f$ and an input specification $\mathcal{X}$, we construct a NAM $f'$ consisting of only one MLP, $f_1 := f$. We start by selecting an arbitrary assignment $\mathbf{x} \in \mathcal{X}$ and checking if $f(\mathbf{x}) < 0$. If this condition is satisfied, we define the weight of the corresponding model as $\beta_1 := 1$ and set the bias term as $\beta_0 := 0$. Furthermore, we initialize the subset $S := \emptyset$. It is easy to see that if there exists a satisfying assignment $\mathbf{z} \in \mathcal{X}$ for which $f(\mathbf{z}) \geq 0$, then $[n] := \{1\}$ is a contrastive reason, since $1 = f'(\mathbf{z}) \neq f'(\mathbf{x}) = 0$ and thus $\emptyset$ is not a sufficient reason for $\langle f', \mathbf{x} \rangle$. In summary, we have constructed an instance of the CSR query where $S$ (the empty set) acts as a sufficient reason for $\langle f', \mathbf{x} \rangle$ if and only if no satisfying assignment exists for the neural network $f$.

For the second case, where $f(\mathbf{x}) \geq 0$, we construct $f'$ in the same way, but this time setting the weight $\beta_1 := -1$, while keeping $\beta_0 := 0$ and $S := \emptyset$. Similarly, we observe that if there exists a satisfying assignment $\mathbf{z} \in \mathcal{X}$ such that $f(\mathbf{z}) \geq 0$, then $[n] := \{1\}$ serves as a contrastive reason, as $0 = f'(\mathbf{z}) \neq f'(\mathbf{x}) = 1$, implying that $\emptyset$ is not a sufficient reason for $\langle f', \mathbf{x} \rangle$. To summarize, we have constructed a CSR query instance where $S$ (the empty set) serves as a sufficient reason for $\langle f', \mathbf{x} \rangle$ if and only if no satisfying assignment exists for the neural network $f$.

This establishes that obtaining the CSR query for a NAM, where $\mathcal{X}$ is defined as either a general discrete or a continuous input domain, is coNP-Hard.

$\square$

**Lemma 15.** *Given a NAM $f$, an input $\mathbf{x} \in \mathbb{F}$, and an integer $k \in \mathbb{N}$, where $f$ is defined over either a continuous input space or a general discrete input space, then the MCR query is NP-Complete.*

*Proof.* **Membership.** For proving membership in NP, we follow the exact same procedure used for proving membership in coNP for the CSR query in Lemma 13. However, instead of verifying that $f(\mathbf{x}) \neq f(\mathbf{x}_S; \mathbf{z}_{\bar{S}})$, we check that $f(\mathbf{x}) = f(\mathbf{x}_S; \mathbf{z}_{\bar{S}})$ while introducing an additional constraint that $|S| \leq k$. Since these conditions can still be determined in polynomial time (with the difference being that we now demonstrate the existence of an instance, rather than the lack of one), this establishes membership in NP.

**Hardness.** We have proven in Lemma 14 that the neural network verification problem for a neural network with one input and one output, where the input domain is either continuous or general discrete, is NP-Hard. We now aim to leverage this hardness result to demonstrate that the MCR query for a NAM defined over either a general discrete or a continuous domain is NP-Hard.

We observe that the hardness proof we derived applies to a specific scenario in which the output specification $\mathcal{Y}$ mandates that the prediction of $f(\mathbf{x})$ must be positive (i.e., $f(\mathbf{x}) \geq 0$). Consequently, we will demonstrate hardness by reducing such an instance—a neural network with a single input and output, a bounded input domain $\mathcal{X}$, where the task is to determine whether $f(\mathbf{x}) \geq 0$ — to the MCR query.

Given a model $f$ and an input specification $\mathcal{X}$, we construct a NAM $f'$ consisting of only one MLP, $f_1 := f$. We start by selecting an arbitrary assignment $\mathbf{x} \in \mathcal{X}$ and checking if $f(\mathbf{x}) < 0$. If this condition is satisfied, we define the weight of the corresponding model as $\beta_1 := 1$ and set the bias term as $\beta_0 := 0$. Furthermore, we set $k := 1$. It is easy to see that if there exists a satisfying assignment $\mathbf{z} \in \mathcal{X}$ for which $f(\mathbf{z}) \geq 0$, then $[n] := \{1\}$ is a contrastive reason, since $1 = f'(\mathbf{z}) \neq f'(\mathbf{x}) = 0$ and thus there exists a contrastive reason of size 1 for $\langle f', \mathbf{x} \rangle$.

For the second case, where $f(\mathbf{x}) \geq 0$, we construct $f'$ in the same way, but this time setting the weight $\beta_1 := -1$, while keeping $\beta_0 := 0$ and, again, setting $k := 1$. Similarly, we observe that if there exists a satisfying assignment $\mathbf{z} \in \mathcal{X}$ such that $f(\mathbf{z}) \geq 0$, then $[n] := \{1\}$ serves as a contrastive reason, as $0 = f'(\mathbf{z}) \neq f'(\mathbf{x}) = 1$, implying that there exists a contrastive reason of size 1. To summarize, we have constructed an MCR query instance where there exists a contrastive reason of size $k := 1$ if and only if there exists a satisfying assignment for the neural network $f$.

This establishes that obtaining the MCR query for a NAM, where $\mathcal{X}$ is defined as either a general discrete or a continuous input domain, is NP-Hard.

$\square$

**Lemma 16.** *Given a NAM $f$, an input $\mathbf{x} \in \mathbb{F}$, and an integer $k \in \mathbb{N}$, where $f$ is defined over either a continuous input space or a general discrete input space, then obtaining the MSR query is coNP-Complete.*

*Proof.* **Hardness.** We have proven in Lemma 14 that the neural network verification problem for a neural network with one input and one output, where the input domain is either continuous or general discrete, is NP-Hard. We can now leverage this hardness result to demonstrate that the MSR query for a NAM defined over either a general discrete or a continuous domain is coNP-Hard.

We, again, can observe that the hardness proof we derived applies to a specific scenario in which the output specification $\mathcal{Y}$ mandates that the prediction of $f(\mathbf{x})$ must be positive (i.e., $f(\mathbf{x}) \geq 0$). Consequently, we will demonstrate hardness by reducing such an instance — a neural network with a single input and output, a bounded input domain $\mathcal{X}$, where the task is to determine whether $f(\mathbf{x}) \geq 0$ — to the (complement of) the MSR query.

Given a model $f$ and an input specification $\mathcal{X}$, we construct a NAM $f'$ consisting of only one MLP, $f_1 := f$. We start by selecting an arbitrary assignment $\mathbf{x} \in \mathcal{X}$ and checking if $f(\mathbf{x}) < 0$. If this condition is satisfied, we define the weight of the corresponding model as $\beta_1 := 1$ and set the bias term as $\beta_0 := 0$. Furthermore, we set $k := 0$. It is easy to see that if there exists a satisfying assignment $\mathbf{z} \in \mathcal{X}$ for which $f(\mathbf{z}) \geq 0$, then $[n] := \{1\}$ is a contrastive reason, since $1 = f'(\mathbf{z}) \neq f'(\mathbf{x}) = 0$ and thus there does not exist a sufficient reason of size 0 for $\langle f', \mathbf{x} \rangle$.

For the second case, where $f(\mathbf{x}) \geq 0$, we construct $f'$ in the same way, but this time setting the weight $\beta_1 := -1$, while keeping $\beta_0 := 0$ and, again setting $k := 0$. Similarly, we observe that if there exists a satisfying assignment $\mathbf{z} \in \mathcal{X}$ such that $f(\mathbf{z}) \geq 0$, then $[n] := \{1\}$ serves as a contrastive reason, as $0 = f'(\mathbf{z}) \neq f'(\mathbf{x}) = 1$, implying that there exists a contrastive reason of size 1, and hence there does not exist a sufficient reason of size 0. To summarize, we have constructed an MSR query instance where there does not exist a sufficient reason of size $k := 0$ if and only if there exists a satisfying assignment for the neural network $f$.

This establishes that obtaining the MSR query for a NAM, where $\mathcal{X}$ is defined as either a general discrete or a continuous input domain, is coNP-Hard. We note that a recent result by [21] strengthens this bound for the optimization variant, showing that the continuous version of the problem is in fact OptP[$\mathcal{O}(\log n)$]-hard under metric reductions (see Section B.5 of the appendix).

Regarding membership, although showing that the problem lies in $\Sigma_2^P$ is straightforward (as the same task is known to be in $\Sigma_2^P$ for general neural networks [16]), we refer the reader to a recent result in [21], which shows that the optimization variant of the problem is in $\text{FP}^{\text{NP}}[\mathcal{O}(n)]$ (see Section B.5 of the appendix). This, in turn, implies that the corresponding decision variant considered here belongs to $\Delta_2^P$.

$\square$

Having completed the proofs for all NAM instances, we now turn to proving the complexity results for EBMs. We will break down the proof into separate lemmas, each handling the complexity argument for one of the explanation types (CSR, NCR, and MSR), starting with the continuous case and then extending to the general discrete case.

**Lemma 17.** *Given an EBM over either a continuous or general discrete input space, then the CSR query is coNP-Complete.*

*Proof.* **Membership.** Although we can directly prove membership for EBMs, we choose to establish it via a reduction to NAMs. Given that the same task is already shown to be coNP-complete for NAMs, and since coNP is closed under polynomial-time reductions, this approach immediately yields

membership in coNP. A key advantage of this proof is that it extends naturally to other queries — such as MCR and MSR — since the same reduction applies to them as well.

We can directly reduce any EBM to a NAM by transforming each boosted tree ensemble $f_i$ into a neural network $f_i'$. This is done by converting each decision tree into a neural network, following the method proposed in [16, 18]. To account for the ensemble structure, we introduce an additional hidden layer that assigns a weight $\phi_j$ to each tree $j$. A standard weighted sum over the outputs then yields a neural network equivalent to the original boosted ensemble. This transformation preserves equivalence for any given input.

**Hardness.** We perform a reduction from the classic tautology problem (*TAUT*) for 3-DNFs (which is coNP-Complete), defined as follows:

---

**TAUT (Tautology)**:
**Input**: A boolean 3-DNF formula $\psi := t_1 \vee t_2 \vee \ldots t_m$.
**Output**: *Yes*, if $\psi$ is a tautology and *No* otherwise.

---

We will begin by proving the hardness for the general discrete input space and then explain how to extend the results to the continuous input space. Given a DNF $\phi := t_1 \vee t_2 \vee \ldots \vee t_m$, we will construct an EBM $f := \langle f_1 \rangle$, i.e., the EBM $f$ is composed of only one boosted tree. It is worth noting that reducing DNFs to boosted trees has been proposed in previous work [63]. However, our situation differs because the boosted tree $f_1$ is defined over only one input feature $x_1$ and operates on some input space $\mathcal{X}$, i.e., $f_1 : \mathcal{X} \to \mathbb{R}$ and $f : \mathcal{X} \to \{0, 1\}$.

We recall that $\phi$ is defined over a set of $n$ literals $X_1, X_2, \ldots, X_n$ and $m$ clauses. Given $\phi$, we will construct a boosted tree model $f_1$ (which will thereby define $f := \langle f_1 \rangle$) and set the general discrete input space $\mathcal{X}_i$ to encompass any possible integer within the range $[0, 2^n - 1]$. Each feature $X_i$ will be associated with its corresponding bit representation in a vector containing $n$ bits.

We note that $f_1$ has one input, $\mathbf{x}_1$, defined over the domain $\mathcal{X}_i$. The model $f_1$ will be an ensemble comprising $m$ decision trees. Each decision tree's input is the feature $\mathbf{x}_1$. Consider a clause $t_i := c_1 \wedge c_2 \wedge c_3$, where each conjunct is a clause within $X_1, \ldots, X_n$ or its negation. For each conjunct $t_i := c_1 \wedge c_2 \wedge c_3$, we will construct a decision tree corresponding to that conjunct.

This decision tree will have three splits, one for each feature $c_1, c_2, c_3$. Assume, without loss of generality, that these features are represented by $X_j$, $X_k$, and $X_l$, where each feature may also be represented negatively as $\bar{X}_j$, $\bar{X}_k$, or $\bar{X}_l$. The splits of the tree will check conditions such as $\mathbf{x}_1 \geq 2^j$, $\mathbf{x}_1 \geq 2^k$, or $\mathbf{x}_1 \geq 2^l$ (representing literals $X_j, X_k, X_l$, respectively), or $\mathbf{x}_1 < 2^j$, $\mathbf{x}_1 < 2^k$, or $\mathbf{x}_1 < 2^l$ (representing literals $\bar{X}_j, \bar{X}_k, \bar{X}_l$, respectively). If all three conditions are satisfied, the tree will output a value of 1; otherwise, it will output 0.

For example, consider the conjunct $t_5 := X_j \wedge \bar{X}_k \wedge X_l$. In this scenario, tree number 5 in the constructed boosted tree ensemble $f_1$ will perform three splits: first, it will check if $\mathbf{x}_i \geq 2^j$, then whether $\mathbf{x}_i < 2^j$, and finally whether $\mathbf{x}_i \geq 2^l$. If all three conditions hold true, the tree will output a value of 1. Otherwise, it will output 0. The $\alpha$ terms (the weights of each tree in $f_1$) will all be set to 1. For the intercept terms of $f$, we will set $\beta_1 = 1$ and $\beta_0 = -1$. This defines the full construction of the EBM $f$. Additionally, we will set $S := \emptyset$. Finally, we construct the input $\mathbf{x}$, which is any integer within the range $[0, 2^n - 1]$, as follows: First, we form a binary vector $\mathbf{x}' \in \{0, 1\}^n$ in the following way: we start by choosing an arbitrary conjunct from $c_1, \ldots, c_m$. For a specific conjunct, we assign the corresponding literals to their respective bits in the vector, while assigning any values to the remaining bits. For instance, consider the clause $c_i := X_1 \wedge \bar{X}_5 \wedge X_8$. In this case, we create a vector $\mathbf{x}'$ where $\mathbf{x}'_1 := 1$, $\mathbf{x}'_5 := 0$, and $\mathbf{x}'_8 := 1$, while the other features in $\mathbf{x}'$ (besides features 1, 5, and 8) are assigned arbitrary values. We then define the input $\mathbf{x}$ as the integer represented by the binary vector $\mathbf{x}'$. In summary, the entire reduction constructs $\langle f, \mathbf{x}, S \rangle$, where $f$ is the constructed EBM defined over the input space $\mathcal{X}_i$ for the single feature $\mathbf{x}_i$ of $f$, $\mathbf{x}$ is the constructed input, and $S := \emptyset$ is the constructed (empty) subset.

We will now prove that $\phi$ is a tautology if and only if $S := \emptyset$ is a sufficient reason for $\langle f, \mathbf{x} \rangle$. From our construction of $\mathbf{x}$ (the integer representation of the binary vector $\mathbf{x}'$, which expands binary bits representing literals from a given conjunct), it follows that for the specific conjunct $c_i$ over which $\mathbf{x}'$ was constructed, the corresponding tree in the ensemble will reach a terminal node with value 1. Consequently, since all other trees output either 0 or 1, and given the intercept term $\beta_0 = -1$,

we have $f(\mathbf{x}) = 1$. Knowing that $f(\mathbf{x}) = 1$, proving that $S = \emptyset$ is a sufficient reason is equivalent to showing that for any $\mathbf{z}$, it holds that $f(\mathbf{x}) = 1 = f(\mathbf{z})$. Conversely, proving that $S = \emptyset$ is not a sufficient reason amounts to demonstrating the existence of some $\mathbf{z}$ such that $f(\mathbf{x}) = 1 \neq f(\mathbf{z}) = 0$.

From our construction, we know that for any assignment of literals $\mathbf{z}' \in \{0, 1\}^n$ over $\phi$, if a conjunct $c_i$ is satisfied, propagating $\mathbf{z}$ (the integer representation of the binary vector $\mathbf{z}'$) through the $i$-th tree of the ensemble will yield a terminal node with value 1. Since the outputs of all other trees are either 0 or 1, it follows that $f_1(\mathbf{z}) \geq 1$. Including the intercept $\beta_0 = -1$, we get $f(\mathbf{z}) = 1 = f(\mathbf{x})$.

On the other hand, if for some assignment $\mathbf{z} \in \{0, 1\}^n$, no conjunct in $\phi$ is satisfied, all trees will reach terminal nodes with value 0, resulting in $f(\mathbf{z}) = 0 \neq 1 = f(\mathbf{x})$. Thus, if there exists an assignment over $\phi$ that evaluates to false (i.e., $\phi$ is not a tautology), there will exist an assignment $\mathbf{z}$ such that $f(\mathbf{z}) = 0 \neq 1 = f(\mathbf{x})$, implying that $S := \emptyset$ is not a sufficient reason for $\langle f, \mathbf{x} \rangle$. Conversely, if no such assignment exists (i.e., $\phi$ is a tautology), then for any $\mathbf{z}$, it holds that $f(\mathbf{x}) = 1 = f(\mathbf{z})$, implying that $\emptyset$ is a sufficient reason. This concludes the proof.

Finally, we note that, for simplicity, we assumed $\mathcal{X}_1$, the domain of the single input $\mathbf{x}_1$, to be a general discrete domain containing all integer values within $[0, 2^n - 1]$. However, the result also holds if $\mathcal{X}_1$ is defined as the entire continuous domain $[0, 2^n - 1]$. Notably, in our construction of the single tree ensemble $f_1$ within the EBM, the decision rule checking whether $\mathbf{x} \geq 2^k$ or $\mathbf{x} < 2^k$ applies equally to the entire continuous domain. Thus, the hardness results extend to the continuous domain as well.

$\square$

**Lemma 18.** *Given an EBM over either a continuous or general discrete input space, then obtaining the MCR query is NP-Complete.*

*Proof.* For *membership*, we can use the same reasoning as in Lemma 17, where the given EBM is reduced to an equivalent NAM. Since NP is closed under polynomial-time reductions, Lemma 18 establishes membership in NP.

For *hardness*, we reduce from the (complement) of the CSR query for EBMs, which we have shown to be coNP-Complete. Specifically, we established coNP-hardness for the CSR query by reducing from the *TAUT* problem to CSR when $S := \emptyset$. Consequently, we prove hardness by reducing from CSR in the special case where $S := \emptyset$. Given an instance $\langle f, \mathbf{x}, S := \emptyset \rangle$, where $f$ is an EBM, we construct an instance $\langle f, \mathbf{x}, k = n \rangle$, where $f$ remains the same EBM, and $k = n$ corresponds to setting $k$ of the MCR query to match the input dimension size. We observe that $S = \emptyset$ is a sufficient reason for $\langle f, \mathbf{x} \rangle$ if and only if $\overline{S}$ is not a contrastive reason for $\langle f, \mathbf{x} \rangle$. Thus, $S = \emptyset$ is a sufficient reason for $\langle f, \mathbf{x} \rangle$ if and only if no contrastive reason of size $n$ exists for $\langle f, \mathbf{x} \rangle$, thereby establishing the reduction.

$\square$

**Lemma 19.** *Given an EBM over either a continuous or general discrete input space, then the MSR query is coNP-Complete.*

*Proof.* For *membership*, the same reasoning as in Lemma 17 and Lemma 18 applies, where the given EBM is reduced to an equivalent NAM. As coNP is closed under polynomial-time reductions, Lemma 17 confirms membership in coNP.

For *hardness*, we reduce from the CSR query for EBMs, which has been shown to be coNP-Complete. Specifically, coNP-hardness for the CSR query was established by reducing from the *TAUT* problem to CSR when $S := \emptyset$. Accordingly, hardness is proved by reducing from CSR in the specific case where $S := \emptyset$. For an instance $\langle f, \mathbf{x}, S := \emptyset \rangle$, where $f$ is an EBM, we construct an instance $\langle f, \mathbf{x}, k = 0 \rangle$, where $f$ remains the same EBM and $k = 0$ corresponds to setting $k$ of the MSR query. It is observed that $S = \emptyset$ if and only if a sufficient reason of size 0 exists for $\langle f, \mathbf{x} \rangle$, thereby establishing the reduction.

$\square$

# J   Proof of Proposition 6

**Proposition 6.** *Given a NAM, a Smooth GAM or an EBM under an enumerable discrete or general discrete setting, the FR query is coNP-Complete.*

*Proof.* **Membership.** Given a GAM $f$ defined over either an enumerable discrete or general discrete input space, and an input feature $i \in [n]$, we can guess two input assignments for feature $i$: $\mathbf{x}_i, \mathbf{z}_i$ such that $\mathbf{x}_i \neq \mathbf{z}_i$. This is achievable due to the discrete nature of the input space: by selecting a single input for each coordinate in the enumerable discrete case, or by choosing a binary vector in the general discrete case. We can additionally guess some assignment for the features $[n] \setminus \{i\}$, which we will denote by $\mathbf{x}'$ (and the correctness holds for the same reasoning). Now, if $f(\mathbf{x}'_{[n]\setminus\{i\}}; \mathbf{x}_i) \neq f(\mathbf{x}'_{[n]\setminus\{i\}}; \mathbf{z}_i)$, then by definition, $i$ is not redundant with respect to $\langle f, i \rangle$, thereby completing the membership proof. We note that this membership result holds regardless of the specific type of GAM — it applies to any GAM, provided that the inference of its components can be performed in polynomial time. Therefore, it directly extends to models such as NAMs, EBMs, and Smooth GAMs.

**Hardness.** The study in [18] established coNP-Hardness for the FR query (referred to there as G-FR) for linear classifiers over a binary input space. Given that this represents a specific instance of the enumerable discrete input setting, where the explicit values $\mathcal{X}_i$ for each feature $\mathbf{x}_i$ are $\{0, 1\}$ (and is therefore also a particular case of the general discrete input setting), the coNP-hardness results apply to these settings as well. We note that linear models are a particular instance of Smooth GAMs, NAMs and EBMs, where each component $f_i$ is taken to be the identity function. Since each of these model types (neural networks, tree ensembles, and piecewise polynomial splines) can trivially be represented as constant identity functions that always produce the same output, the hardness result naturally extends to EBMs, Smooth GAMs, and NAMs.

$\square$

# K   Proof of Proposition 7

**Proposition 7.** *Given NAMs and EBMs over discrete or continuous settings, computing SHAP for regression tasks is #P-Complete. Moreover, given* any *GAM over an enumerable discrete, discrete or continuous setting - computing the CC and SHAP queries for classification are #P-Complete.*

*Proof.* We will divide the upcoming proof into two distinct lemmas.

**Lemma 20.** *Given NAMs and EBMs over discrete or continuous settings, computing SHAP for regression tasks is #P-Hard.*

*Proof.* We will start by proving the results for NAMs. First, similarly to the proof in Lemma 5, we will assume we are dealing with a single input-output neural network, and hence, hardness for this scenario will directly apply to full NAMs. We base the reduction based on the results obtained by [7] and [102] which showed that computing SHAP for a model $f$ and an input $\mathbf{x}$ is at least as hard as the model counting problem for $f$, under the uniform distribution assumption (and this clearly shows hardness for the more general distribution setting). CNF formulas can be reduced in polynomial time to neural networks, as was shown by multiple studies [16, 69, 95, 18] (and this claim actually holds even for boolean circuits [16]). As shown by [69] and later refined by [95], a neural network defined over a continuous domain can be reduced to the discrete case, thereby transferring the hardness results to the continuous setting as well.

In Lemma 5, we showed that a neural network verification problem for a full network can be reduced to one involving a single input-output pair. Since the former is known to be hard via a reduction from CNF-SAT [69, 95], this implies that the counting version of CNF-SAT can likewise be reduced to counting the number of satisfying assignments in a single input-output neural network. This yields a parsimonious reduction, establishing #P-hardness. As EBMs can be reduced from NAMs (as we show in Lemma 5), this hardness result extends to EBMs as well.

$\square$

**Lemma 21.** *Given any* GAM *over an enumerable discrete, discrete or continuous setting - computing the CC and SHAP queries for classification are #P-Hard.*

*Proof.* Hardness results for GAMs persist because they apply even to *linear* classification models, which are a special case of GAMs. Consequently, the hardness remains for GAMs. Specifically, for the CC query, this result was established by [16], and for the SHAP query, it was demonstrated by [102]. Linear models can be viewed as a special case of Smooth GAMs, NAMs, and EBMs, where

each component function $f_i$ is simply the identity function. Since neural networks, tree ensembles, and piecewise polynomial splines can trivially replicate this behavior by acting as constant identity functions that yield a fixed output, the hardness result directly extends to EBMs, Smooth GAMs, and NAMs.

□

