# OpenReview forum: "Additive Models Explained: A Computational Complexity Approach"
_NeurIPS.cc/2025/Conference — NeurIPS 2025 poster_

### Official Review · Reviewer_gVvN · 2025-06-23

**Clarity:** 3
**Significance:** 2
**Originality:** 2
**Rating:** 4
**Confidence:** 2

**Summary:**

**Because I am not an expert in GAMs and even less in computational‑complexity theory, I attach a low confidence score to all judgments below.**

The paper studies how hard it is provably to compute several popular explanation types for GAMs.
Across three axes (input-space discreteness, component model class, and regression vs. classification task), the authors present polynomial-time algorithms, NP/coNP hardness, and #P hardness results, summarized in Table 1. Key insights include
(i) complexity often hinges on whether the input domain is enumerable, discrete, or continuous;
(ii) splines, neural additive models (NAMs), and explainable boosting machines (EBMs) yield different hardness profiles; and
(iii) SHAP is tractable for regression but #P-hard for classification.

**Questions:**

- Q1. How do these theoretical complexity results translate to practical computational differences on real datasets of typical sizes?
- Q2. Are there good polynomial-time approximation algorithms for the cases proven to be intractable?
- Q3. How common are the different input domain scenarios in real-world applications of GAMs?
- Q4. Could the analysis be extended to other explanation methods commonly used in practice (integrated gradient, banzhaff indices, sobol' indices) ?

**Ethical Concerns:**

["NO or VERY MINOR ethics concerns only"]

**Final Justification:**

I keep my borderline-accept rating. The rebuttal provides helpful clarifications and outlines possible extensions, but the work still lacks empirical grounding to show whether the complexity distinctions have practical impact. Even small experiments or runtime comparisons would help connect theory to application. I am not an expert in GAMs or computational complexity, so I give low weight to my judgment here, but I remain unconvinced of the practical utility despite the theoretical solidity.

**Limitations:**

yes

**Paper Formatting Concerns:**

no formatting concerns

**Quality:**

3

**Strengths And Weaknesses:**

**Strenghts**

- The framework / splitting of the research into (input, component, task) make the reading of the article pleasant and also more easily accessible, thank you.
- The authors occasionally extract intuitive explanations, for example, why classification SHAP becomes hard once a step function breaks linearity, which helps readers who lack deep complexity background. Their pseudo-polynomial algorithms show the results are not purely negative; they identify practical levers (weight precision) to regain tractability.

**Weaknesses**

- M1. My main concern is about limited practical relevance: While theoretically interesting, many of the complexity distinctions may not matter in practice. Can you elaborate or help translate your finding into any actionable insight ?
- M2. Type of explanations: The analysis relies heavily on specific mathematical definitions of explanation types that may not align with how practitioners actually use or interpret GAM explanations (maximally activating example, gradient...). The theoretical definitions might be too narrow for practical explainability.
- M3. Related to M1, Missing empirical validation: As a purely theoretical work, there's no validation of whether the complexity distinctions translate to meaningful differences in real-world computational performance or whether the theoretical worst-case scenarios occur in practice.

Now for the minors weaknesses,
- m1. dense presentation: The paper is quite technical and may be difficult to access for researchers who aren't specialists in computational complexity theory, potentially limiting its impact.
- m2. limited discussion of approximation algorithms: While the paper establishes intractability results, it doesn't explore whether good polynomial-time approximation algorithms might exist for the hard cases.
- m3. typo: “continous”
- m4. lines 221–230 "often relevant for tabular data"; a citation to a real tabular application using enumerable GAMs would help.

---

> ### Author Rebuttal · Authors · 2025-07-30
>
> We thank the reviewer for their valuable and constructive feedback and for acknowledging the importance of our work. See our detailed response below.
>
> **Actionable insights and practicality of the theoretical results**
>
> We thank the reviewer for recognizing the theoretical significance of our work and for their appreciation of both the novel results and their presentation. While the paper is indeed theoretical, we believe it offers many important practical insights for GAM practitioners. In particular, our study introduces a broad set of *efficient algorithms* for computing widely used and well-established explanation types from the literature — including minimum sufficient reasons, minimum contrastive reasons, feature redundancy detection, and exact SHAP attributions.
>
> As demonstrated in Table 1, we prove that many of these computations are solvable in polynomial time across various GAM configurations. Moreover, we will make it clearer in the final version that our tractable algorithms are not only polynomial but also *linear* in the size of the GAM, emphasizing their practical usability.
>
> For example, in loan applications, a practitioner may seek a minimal feature subset that guarantees approval (a minimum sufficient reason) or the smallest change needed to reverse an undesired rejection (a minimum contrastive reason). In healthcare, identifying redundant features, like blood pressure being irrelevant to any prediction, can be critical for making decisions. SHAP values, which assign exact importance scores, are widely used in fields like tabular data and computer vision. Our results pinpoint when such explanations can be computed efficiently, guiding practitioners in configuring a GAM accordingly and applying our algorithms.
>
> It is also important to mention that even in cases where we establish intractability, we sometimes provide *pseudo*-polynomial algorithms that become efficient when the GAM’s coefficients are quantized, that is, restricted to limited precision. While these are less broadly applicable than our polynomial-time algorithms, they still offer a practical solution for users working under reasonable constraints. In particular, a practitioner can configure a GAM with quantized weights and still efficiently compute the desired explanation using our algorithm.
>
> Moreover, we believe our intractability results (e.g., NP-hardness, #P-hardness) also offer practical value by identifying which explanations are computationally feasible and which are inherently hard. One particularly important insight is that the *input domain* plays a crucial role in determining tractability — a dependency that has not been found on other popular ML model classes. This observation can guide practitioners in configuring GAMs (in terms of both input encoding and component selection) to enable efficient explanation generation.
>
> Finally, we fully agree with the reviewer that, while our work establishes hardness results in certain settings, investigating heuristic approaches to overcome these computational challenges is a valuable and promising avenue for future research. Our *tractable* algorithms are tailored to specific configurations (e.g., particular component types or input domains) and thus cannot be directly extended to the intractable cases. Nonetheless, we believe that these algorithms, together with the computational boundaries they outline, can offer a strong foundation for the development of heuristics that may bypass some of the established hardness barriers in generating explanations. Designing such heuristics would likely require significant algorithmic design and constitutes a meaningful research challenge in its own right. We will highlight this as a central direction for future work and believe our contributions provide a strong basis for initiating such efforts.
>
> Overall, we thank the reviewer for highlighting these important points and will incorporate a thorough discussion of them in the final version.
>
>
> **Possible extensions to additional explanation forms**
>
> We agree with the reviewer that, while our framework captures a broad range of widely-used explanation types, other explanation notions are certainly worth exploring. Indeed, many of our proof techniques can be extended to additional forms. For example, *Banzhaf values*, that were mentioned by the reviewer, share key complexity-theoretic properties with Shapley values, and some of our Shapley-based proof techniques (e.g., using the linearity axiom) naturally carry over.
>
> Similarly, when computing minimum sufficient or contrastive reasons, our algorithm identifies the minimum or maximum viable values for each GAM component and ranks them accordingly. As a result, a subroutine of this algorithm can also be used to generate *maximally activating examples* — an explanation type noted by the reviewer. In the case of SHAP explanations, which require expected value computations, a subroutine of that algorithm can be extended to support *variance*-based approaches such as *Sobol indices*, which also rely on expectation calculations. More broadly, many explanation methods involve identifying critical values, ranking components, or computing expectations — all of which align with the core procedures used in our algorithms.
>
> In summary, we agree that extending the analysis to additional explanation types is a promising direction for future work. While we believe that our current focus captures a rich and representative subset of widely-used and practically relevant explanation methods, we also believe that the breadth of algorithmic and proof techniques developed in this work provides a strong theoretical foundation for analyzing further explanation forms in future work. We thank the reviewer for raising this point and we plan to incorporate a thorough discussion of this matter in our final version.
>
> **Approximation and circumvention schemes for intractable results**
>
> While our work offers a broad characterization of both tractable and intractable cases for explanation computation across a variety of settings, it also identifies two particularly promising strategies for circumventing intractability in otherwise hard cases.
>
> The first strategy involves *pseudo*-polynomial algorithms. These show that, although the general problem is intractable, assuming quantized GAM coefficients (i.e., bounded precision) renders the problem efficiently solvable.
>
> The second strategy builds on our findings regarding the role of the input domain in shaping complexity. In some cases, reconfiguring the input space provides a practical workaround. For instance, while explanation computation may be intractable over general discrete or continuous domains, it becomes tractable when the input is enumerable (or vice versa for other explanations). This suggests that input quantization, or a different transformation technique, can enable efficient computation.
>
> We also agree that additional complexity circumvention strategies represent an important avenue for future work. For example, can we design fully polynomial-time approximation schemes (FPTAS) or randomized schemes (FPRAS) to approximate the *size* of sufficient or contrastive explanations? Furthermore, are there structural properties, beyond input representation and coefficient precision, that could make intractable explanation problems tractable?
>
> We will incorporate these open directions into our discussion of future work and highlight the two strategies above more prominently in the final version. We thank the reviewer for bringing this up.
>
> **The common use of different input domains in GAMs**
>
> Yes, all the input domain types we analyze are commonly used in practice when working with GAMs and typically depend on how the underlying features are represented in the data ([1–3]). For instance, in the standard tabular task of loan prediction, features like *age* or *monthly income* can often be modeled as continuous, as they represent values over a continuous range. In contrast, features such as *occupation* (“teacher”, “engineer”, “unemployed”), *gender*, or *loan purpose* (“business”, “wedding”, “education”, etc.) are typically modeled as discrete inputs with a fixed and enumerable set of options. However, other features like *user name* or *ID* are also discrete, yet effectively non-enumerable due to the vast number of potential unique values.
>
> It is also worth noting that different input representations can often be transformed into one another. In GAMs, discretization is a common practice (e.g., [1,3]). For example, in our loan application example, continuous inputs like *age* or *salary* can be left as-is, discretized into broad (non-enumerable) buckets, or even quantized into a small enumerable set of categories. Our complexity results highlight how the input domain has a substantial impact on the feasibility of computing various explanation types, and this suggests practical mitigation strategies: by adjusting how input features are represented, practitioners can potentially improve the efficiency of explanation computation. We appreciate the reviewer’s point on this matter and plan to expand on this discussion in the final version of the paper.
>
>
> **Presentation and other minor suggestions**
>
> We thank the reviewer for these helpful suggestions! Following these remarks, we will use the additional space in the final version to provide more background and intuitive explanations regarding the computational complexity aspects of the paper. We will also fix the noted typo and include the missing reference. We appreciate your careful reading!
>
> [1] Intelligible Models for HealthCare: Predicting Pneumonia Risk and Hospital 30-day Readmission (Crauna et al., KDD 2015)
>
> [2] Axiomatic Interpretability for Multiclass Additive Models (Zhang et al., KDD 2019)
>
> [3] Accuracy, Interpretability, and Differential Privacy via Explainable Boosting (Nori et al., ICML 2021)

---

> > ### Comment · Reviewer_gVvN · 2025-08-03
> > **Thank you for your response.**
> >
> > Thank you for the detailed clarifications. I appreciate the response and the outlined extensions.
> >
> > That said, I still think the paper would benefit from **stronger connection to practical use**. The theoretical results are interesting, but without experiments or concrete examples, it remains unclear whether these complexity differences matter in real applications. Even small empirical validations or runtime comparisons could help ground the claims.
> >
> > In my view, the paper is solid as a theoretical contribution, but could be more impactful if it engaged more directly with practical scenarios.

---

> > > ### Author Response · Authors · 2025-08-04
> > >
> > > We thank the reviewer for their response and are glad they appreciated the outlined extensions, and for acknowledging the interesting and solid theoretical contributions of our work, as well as their overall support for the paper.
> > >
> > > We do note that we believe that presenting significantly tractable algorithms, running in linear time regardless of dataset size, for computing highly desirable explanation definitions in certain GAM settings, alongside proving substantial hardness results for others (e.g., NP-Hard, or even worse), has strong practical value. These results serve as clear guidance on when explanation computations are tractable and implementable in practice, and when they are fundamentally infeasible — highlighting the need to pursue alternative strategies, such as developing different forms of heuristic approaches, which may require substantial innovation in their own right.
> > >
> > > Moreover, our results also offer clear practical directions for addressing intractability in GAMs through two especially relevant strategies:  (1) *input transformations* (e.g., discretization), which is widely used in practice in GAMs and justified by the substantial complexity differences we identify across input domains, and (2) *quantization* of GAM coefficients, motivated by our pseudo-polynomial tractability results. Both approaches significantly improve computational feasibility, offering valuable and actionable insights for practitioners working with GAMs.
> > >
> > > While we fully agree that empirical studies play a vital role, we also recognize that NeurIPS has a strong tradition of valuing theoretical contributions, many of which have ultimately led to significant practical advances. In this context, we believe the theoretical focus of our work is a strength, particularly given the clear practical relevance we have demonstrated.
> > >
> > > We see significant potential for future work building on our results, such as implementing variants of the proposed algorithms, designing heuristics for intractable settings, and empirically investigating input discretization and quantization strategies, which we have proven to be critical. However, due to the space needed to present the breadth of our many theoretical contributions, we were unable to include such experiments within the scope of this paper. We view these empirical avenues as valuable but separate directions that warrant thorough exploration on their own. That said, we believe the theoretical foundations established here are self-contained and well-suited to guide and inform these future empirical efforts.
> > >
> > > We thank the reviewer once again for their valuable feedback and insightful questions. Building on these important points, we will ensure that the final version includes a more comprehensive discussion of these aspects.

---

### Official Review · Reviewer_mpma · 2025-06-27

**Clarity:** 2
**Significance:** 3
**Originality:** 3
**Rating:** 5
**Confidence:** 3

**Summary:**

Generalized additive models are a popular class of glassbox machine learning methods which are easily interpretable through the shape functions of each feature which detail each feature's contribution to the output. However, in order to make actionable decisions based on generalized additive models one requires further explainability methods such as minimum sufficient reason. The present paper presents computational complexity results for several explainability methods which depends on the input domain and the model class of the additive components and problem type (regression or classification). This is a theory paper without experiments.

**Questions:**

-

**Ethical Concerns:**

["NO or VERY MINOR ethics concerns only"]

**Final Justification:**

The authors addressed all of my concerns, I think they made the paper both more accessible to readers unfamiliar with complexity theory as well as more rounded by including a discussion on concurvity.

**Limitations:**

-

**Paper Formatting Concerns:**

-

**Quality:**

3

**Strengths And Weaknesses:**

I am unfamiliar with complexity theory and hence did not check the math.

### Strengths:
- The paper is well-structured, polished and easy to read. I would only suggest to add more intuition here and there as mentioned below (minor).
- All theoretical contributions are well summarized in section 4 and 5.

### Weaknesses:
- Could you add a discussion on how the complexities you derived relate to concurvity [see e.g. Kovács (2024)]? Would stronger concurvity lead to higher computational complexity in some way?
- Can you add a discussion relating your results to a reader (like myself) who is unfamiliar with complexity theory? What could be actionable consequences of your findings for a data scientist?

#### (Minor):
- I think it would help a reader new to the field to add intuition to the explanation types in section 3.3, for example mention a real-world scenario in which this explanation might be useful. For example for minimum sufficient reason it could be to ensure that a loan application is approved only based on certain attributes and that the decision is not based on others. For minimum change required it could be to determine the minimum change in a feature required to go get a customer from non-buying to buying.

### References:
- Kovács, L. Feature selection algorithms in generalized additive models under concurvity. Comput Stat 39, 461–493 (2024). https://doi.org/10.1007/s00180-022-01292-7

---

> ### Author Rebuttal · Authors · 2025-07-30
>
> We thank the reviewer for their valuable and constructive feedback and for acknowledging the significance of our work. See our detailed response below.
>
> **Actionable consequences of complexity results**
>
> We thank the reviewer for acknowledging the theoretical contributions of our work. Although the focus of our paper is theoretical, we believe it offers many valuable practical insights for practitioners working with GAMs. Notably, our study introduces a wide range of *efficient algorithms* for computing several widely used and well-regarded explanation types from the literature — such as minimum sufficient reasons, minimum contrastive reasons, identifying feature redundancies, and exact SHAP explanations. We present algorithms in this paper demonstrating that, for many configuration choices (as outlined in Table 1), these computations can be solved in polynomial time. Moreover, we will clarify more explicitly in the paper that our algorithms are not just polynomial, but are *linear* in the size of the GAM, highlighting their practical applicability.
>
> Moreover, even in cases where we establish intractability, we sometimes provide *pseudo*-polynomial algorithms. These algorithms become highly efficient when the GAM’s coefficients are quantized (i.e., have limited precision), unlike the even more desirable general tractable cases, which do not rely on such assumptions. This presents an additional practical strategy for practitioners, who can quantize a GAM’s precision weights to facilitate more efficient explanation computation.
>
>
> Finally, we believe that our many *intractability* proofs (e.g., NP-hardness, #P-hardness) also offer critical practical guidance by delineating which types of explanations are computationally feasible and which are not. One especially relevant insight is that the modeling of the *input space* has a critical impact on tractability — unlike other ML models where such a dependency has not been observed. This has practical implications for how practitioners configure GAMs (in terms of the choice of input domain and component types) when aiming to obtain specific forms of explanations efficiently. For example, practitioners may choose to reconfigure the input space, particularly by applying transformations such as discretization, quantization, or switching between discrete and continuous domains (and vice versa), to enable efficient explanation computation, depending on the specific explanation type of interest.
>
>
> We thank the reviewer for highlighting this important point and will discuss these aspects in detail in the final version of the paper.
>
> **Relation of complexity results to the concurvity of GAMs**
>
> We thank the reviewer for raising this interesting point regarding concurvity in GAMs, particularly in connection with the work by Kovács et al. Our analysis does not rely on any structural assumptions such as low concurvity — all of our tractability results (i.e., polynomial-time algorithms) hold for *any* GAM, regardless of the level of concurvity, as long as the configuration of input domain, component type, and explanation type is fixed. In other words, the efficiency of our algorithms remains unchanged whether the model exhibits high or low concurvity.
>
> That said, while our intractability results (e.g., NP-hardness, #P-hardness) hold in general, introducing additional structural assumptions on the GAM could, in some cases, reduce the complexity of computing explanations — depending on the specific assumption, explanation type, input domain, and component models. In this regard, we agree that concurvity is a particularly interesting parameter to investigate. For example, consider two extreme cases: in the case of very high concurvity, where the smooth components are highly correlated and effectively behave similarly to even a single spline, the GAM may resemble a simpler function class, which could make certain explanation tasks easier. Conversely, in scenarios of very low concurvity, especially when considering explanation types that depend on the interaction between features, the high separability of components might also lead to computational advantages. Naturally, the precise implications depend on the exact formal definition of concurvity and the specific configuration of the input domain, explanation task, and component models. We agree that this is a valuable and interesting direction, and we will include a discussion of potential structural assumptions worth exploring, with particular focus on concurvity and its connection to Kovács et al., in the final version.
>
>
> **Real-world scenarios of explanation definitions**
>
> We thank the reviewer for this valuable suggestion and fully agree with its importance. Following this, we plan to include in our final version real-world examples to illustrate each of the explanation definitions presented in our work. For instance, in the context of loan applications, a user may be interested in identifying a minimal subset of input features that alone suffice to predict a loan approval or denial (a minimum sufficient reason), or in providing an actionable explanation for what specific minimal change would be required to flip the application decision from rejection to approval (a minimum contrastive reason). Feature redundancy identification can be particularly relevant in domains like healthcare, where recognizing that a certain criterion, such as blood pressure, is redundant can have significant implications for diagnosis or treatment decisions. SHAP attributions, which assign additive importance scores to features, can be widely applicable across tasks involving tabular data, computer vision, and more. We agree that including such examples will help clarify the explanation types and improve the paper’s accessibility, and we will incorporate them into the final version.

---

> > ### Comment · Reviewer_mpma · 2025-08-02
> >
> > I thank the authors for their thoughtful rebuttal. All of my concerns were addressed, in particular I like the added discussion on concurvity which often occurs in practice, because of multicollinearity in the features.
> > I am increasing my score to accept.

---

### Official Review · Reviewer_gmVs · 2025-07-03

**Clarity:** 2
**Significance:** 2
**Originality:** 3
**Rating:** 4
**Confidence:** 4

**Summary:**

This paper analyzes the computational complexity of generating explanations for generalized additive models (GAMs), including spline-based GAMs, neural additive models (NAMs), and explainable boosting machines (EBMs). It formalizes the hardness of producing four types of explanations across different types of input domains (enumerable discrete, general discrete, and continuous). One of the most interesting insights is that, unlike for other model classes (e.g., decision trees or neural networks), the tractability of explanations in GAMs depends on the nature of the input domain. In a somewhat counterintuitive result, certain explanation types are shown to be strictly easier in continuous input spaces than in discrete ones. The paper contributes to the theory of explainable AI by highlighting structural distinctions that affect the computational feasibility of explanation, even for models typically viewed as interpretable.

**Questions:**

- Are any of the intractable explanation queries known to admit efficient approximation schemes or heuristics in practice? If so, a discussion of this would improve the practical significance of the work.
- Do the complexity results extend to GAMs with pairwise or higher-order interactions, such as those implemented in EBMs with interaction terms?
- How do the theoretical results relate to known empirical inconsistencies in GAM explanations (e.g., contradictory attributions across model classes [Chang KDD 2021])? If the authors can connect their results to these empirical observations, it would strengthen the paper’s impact.

**Ethical Concerns:**

["NO or VERY MINOR ethics concerns only"]

**Final Justification:**

Thank you for your detailed response to my and other reviewer comments.

I appreciate the clarifications regarding the role of input domain in explanation complexity and the potential extensions to interaction terms. Your comments on approximation strategies and prior empirical work help clarify the broader context of the results.

That said, most of the rebuttal focuses on future revisions rather than new contributions or clarifications within the current submission. The points raised around practical relevance, empirical connections, and approximation techniques are still left to the domain of future work.

I continue to view the paper as a technically sound theoretical contribution with an interesting, though perhaps unsurprising, result about domain-sensitive explanation complexity in GAMs. However, the practical utility remains limited, and I am keeping my score unchanged at a borderline accept.

**Limitations:**

Yes.

**Quality:**

3

**Strengths And Weaknesses:**

Strengths:
- A welcome contribution to formal explainable AI, in a field often dominated by heuristics and empirical metrics.
- Provides a clean, well-structured taxonomy of explanation types and complexity results.
- The finding that explanation complexity for GAMs is sensitive to the input domain is genuinely interesting and distinguishes additive models from other common model families.

Weaknesses:
- While technically correct, many of the hardness results (e.g., for NAMs or EBMs) are not surprising given the high capacity and unbounded VC dimension of these models.
- No empirical demonstrations or approximation strategies are provided, and the practical relevance of the theoretical results is limited.
- The paper does not engage with prior empirical work on inconsistencies in GAM explanations, which could help contextualize the formal findings.
- The writing is a bit wordy, and repeatedly frames results as “surprising,” even in cases where the conclusions could be understood directly from known properties. This undermines the impact of the more novel insights.

---

> ### Author Rebuttal · Authors · 2025-07-30
>
> We thank the reviewer for their valuable and constructive feedback and for acknowledging the importance of our work. See our detailed response below.
>
> **Better clarifying the unexpectedness of certain findings**
>
> We thank the reviewer for raising this point. As the reviewer noted, some of the findings in our work may indeed seem surprising. For instance, the sharp complexity differences observed in additive models depending on the input domain contrast with the behavior seen in other ML models. Moreover, the fact that complexity in GAMs varies *inconsistently* across input domains, with continuous domains sometimes making explanation computation harder and other times easier, also stands out as unexpected.
>
> That said, we agree that certain findings may indeed seem less surprising — such as the general observation that computing explanations for GAMs can, in *some* configurations, be computationally challenging. As the reviewer correctly pointed out, this result can perhaps align with intuitive expectations based on the unbounded VC dimension of GAMs in some settings. Still, even in these seemingly more expected cases, our results reveal significantly *diverse* complexity distinctions that depend on the specific configuration of component model type, input domain, and explanation type.
>
> Overall, we certainly agree with the reviewer’s suggestion that we should more carefully qualify terms like “surprising” or “unexpected” and explain their exact meaning. Our intention was to highlight results that deviate from typical patterns, such as the significantly different complexity behavior of additive models compared to other ML models or the inconsistent role of the input domain. Nonetheless, we agree that such terms can be subjective, and we will revise the final version to clarify their usage and provide more precise context.
>
> **Extension to higher-order interaction GAMs**
>
> We thank the reviewer for this excellent question. We agree that exploring the complexity of explanations under higher-order interactions is a very interesting direction. The answer depends heavily on how such interactions are defined — for instance, whether we allow a fixed number $k$ of interaction terms or an unbounded number of interactions up to degree $k$ — as well as on the specific configuration of explanation type, component model, and input domain, which our work shows significantly affect complexity. In some cases (e.g., for SHAP or feature redundancy identification), adding a small number of interactions may be handled with straightforward extensions of our existing algorithms. In contrast, our algorithms for computing explanations like sufficient or contrastive reasons, which rely on sorting features by importance, may become more complicated. In general, while some explanation types could extend naturally to handle interactions, others may exhibit new complexity behaviors that warrant deeper investigation. We see this as a very interesting and fundamental direction for future research, and in the final version, we plan to discuss it further, providing complexity extensions for some simple cases and framing the broader “price of interaction” as a central open direction for a full and thorough investigation in future work.
>
> **Relation to prior work on the interpretation and inconsistencies of GAMs**
>
> We believe that our work, as highlighted by the reviewer as well as other reviewers, offers a unique and previously unexplored perspective on the explainability of GAMs by focusing on the computational aspects of generating explanations. That said, we recognize that some of our findings indeed relate to existing ideas in the literature concerning interpretability and inconsistencies in GAMs, and we agree with the reviewer that improving the discussion of these connections would further strengthen our work.
>
> One clear connection to prior work lies in the observation that GAMs with more complex or non-linear component models are often harder to interpret than those built from simpler components. This observation has been noted in several prior studies (e.g., [1-3]) and aligns with some of our own findings (see Table 1 and Section 5.2), which show that explanation computation can indeed become more difficult for GAMs with more expressive components. However, our results also reveal that this is not a universal rule — the complexity of obtaining explanations depends heavily on both the type of explanation and the nature of the input domain. We hence believe that this new perspective offers a more nuanced and rigorous understanding of when and why some of these interpretability challenges arise.
>
> A second connection to prior work lies in the widely held assumption that additive models are inherently more interpretable than their non-additive counterparts, which is a central motivation for their use. Although this assumption is often treated as self-evident, our work offers a new computational perspective: we demonstrate that transitioning from non-additive models (such as neural networks) to additive ones (such as NAMs) can indeed simplify the computation of explanations. However, we find that this advantage is also not universal; it depends on the specific pairing of input domain types and explanation forms, which once again highlights the value of this computational angle.
>
> Finally, we agree with the reviewer that incorporating a discussion of inconsistencies, such as those identified by Chang et al. (KDD, 2021) and related work, would enhance our paper, and we intend to include this in the final version. These studies underscore the importance of *data fidelity* in interpreting GAMs, demonstrating that models with similar predictive performance but different architectures (e.g., splines vs. EBMs) can vary significantly in their bias-variance tradeoffs. This points to the *high sensitivity* of learned GAM functions to both the underlying data distribution and the chosen component models. Although this prior work is empirical and differs from our computational perspective, both lines of research highlight a shared insight: the component structure and input representation in GAMs have a crucial role in shaping the model’s functional form — which, in turn, directly influences both its interpretability and the complexity of generating explanations. We thank the reviewer for highlighting this additional perspective. We plan to incorporate this discussion into our final version, and we agree it will help strengthen our work’s connection to broader discussions in the literature.
>
>
> **Approximation and circumvention schemes for intractable results**
>
> We thank the reviewer for raising this point. While our work presents a broad range of results characterizing both strict tractability and intractability of explanation computations across various settings, it also highlights two particularly promising avenues for circumventing intractability in otherwise hard cases.
>
> The first avenue involves *pseudo*-polynomial algorithms. These results demonstrate that although the general problem may be intractable, if one assumes that the coefficients of the GAM are quantized (i.e., represented with bounded precision), the problem becomes computationally efficient to solve.
>
> The second avenue stems from our findings on how the input domain influences complexity. In some cases, changing the input configuration offers a practical workaround to intractability. For example, we show that while explanation computation may be intractable over a continuous or general discrete input space, it becomes tractable when the domain is an enumerable discrete set (or vice versa) — suggesting that input quantization, or other input transformation techniques, can enable efficient computation.
>
> We also agree with the reviewer that further exploration of additional complexity circumvention strategies is an important direction for future work. For instance, can we develop fully polynomial-time approximation schemes (FPTAS) or fully polynomial-time randomized approximation schemes (FPRAS) that approximate the *size* of sufficient or contrastive explanations? Additionally, are there other structural parameters, beyond input representation and coefficient precision, that could render intractable explanation tasks tractable?
>
> We will make sure to include these open directions as part of our discussion of future work and will also better emphasize the above two circumvention strategies in the final version.
>
> [1] Interpretable Machine Learning (Molnar et al.)
>
> [2] How Interpretable and Trustworthy are GAMs? (Chang et al., KDD 2021)
>
> [3] Neural Additive Models: Interpretable Machine Learning with Neural Nets (Agarwal et al., Neurips 2021)

---

> > ### Comment · Reviewer_gmVs · 2025-08-07
> >
> > Thank you for your detailed response to my and other reviewer comments.
> >
> > I appreciate the clarifications regarding the role of input domain in explanation complexity and the potential extensions to interaction terms. Your comments on approximation strategies and prior empirical work help clarify the broader context of the results.
> >
> > That said, most of the rebuttal focuses on future revisions rather than new contributions or clarifications within the current submission. The points raised around practical relevance, empirical connections, and approximation techniques are still left to the domain of future work.
> >
> > I continue to view the paper as a technically sound theoretical contribution with an interesting, though perhaps unsurprising, result about domain-sensitive explanation complexity in GAMs. However, the practical utility remains limited, and I am keeping my score unchanged.

---

> > > ### Author Response · Authors · 2025-08-08
> > >
> > > We thank the reviewer for their thoughtful response and valuable feedback. We are glad our rebuttal and clarifications were helpful, and we appreciate the recognition of our interesting theoretical contributions and the reviewer’s overall support of our work.
> > >
> > > We would like to emphasize that several central points from our rebuttal are easy to incorporate into the final version. In particular, the two main approximation circumvention techniques, (1) input domain transformations and (2) GAM weight quantization, follow directly from our complexity results and can be straightforwardly highlighted. The same applies to the reviewer’s suggestion to discuss additional prior empirical work, which we agree would strengthen the connection to existing literature.
> > >
> > > We would also like to clarify a point that we fear may have led to a small misunderstanding in our previous response, particularly given the reviewer’s mention of the “perhaps unsurprising results regarding the input domain”, which we suspect perhaps relates to our discussion of the VC dimension of GAMs. We highlight: while it is true that the VC dimension of a GAM can be unbounded in certain configurations, this *does not directly imply* that explanation computation becomes hard. For instance, decision trees also have unbounded VC dimensions, yet many explanations we prove to be (sometimes) intractable for GAMs, such as (1) minimum contrastive subsets, (2) Shapley values, and (3) feature redundancy detection, *can always* be computed for decision trees in polynomial (even linear) time. This shows that an unbounded VC dimension *does not* inherently lead to computational hardness of explanation computations. In fact, even when the VC dimension *is* bounded (e.g., in Smooth GAMs), we prove that many explanation tasks remain hard. In short, while an unbounded VC dimension may intuitively “hint” on an increased complexity in some cases, it is far from a definitive indicator. The results we provide for GAMs in this context are highly diverse.
> > >
> > > Moreover, the unexpected role of the input domain in shaping complexity is further evident in two additional ways: (1) GAMs show substantial input sensitivity *in stark contrast* to many other models like decision trees, tree ensembles, and neural networks, where this does not occur; and (2) this sensitivity is *inconsistent*, as continuous domains can sometimes make explanation computation in GAMs harder, and other times easier.
> > >
> > > We will make sure to better emphasize these points, along with the other aspects mentioned, in the final version. Many thanks again to the reviewer for the valuable remarks and suggestions!

---

### Official Review · Reviewer_zumg · 2025-07-03

**Clarity:** 3
**Significance:** 3
**Originality:** 3
**Rating:** 5
**Confidence:** 2

**Summary:**

This paper discusses Generalized Additive Models from complexity theoretic perspective, challenging the hypothesis that obtaining meaningful explanations for GAMs could be performed efficiently and would not be computationally infeasible.

The analysis is outlined in three dimensions. component model type, type of explanations, and input domain setting. There are several key findings that the analysis shows, including the heavy impact from input domain and component type on the GAM explanations and the hardness of explanations for classification task from SHAP model. In addition, depending on the explanation types and input domain,  some black-box model like neural networks are harder to explain compared with the additive models like NAM.

The paper further details each finding along with proof in the appendix.

**Questions:**

1. Why is the LIME not included in the selection?

**Ethical Concerns:**

["NO or VERY MINOR ethics concerns only"]

**Final Justification:**

I thank the authors for the detailed explanations. I changed my score accordingly. However, as mentioned, I am not an expert in this the complexity analysis. Thus, I would keep my confidence score.

**Limitations:**

None.

**Paper Formatting Concerns:**

None.

**Quality:**

3

**Strengths And Weaknesses:**

Strengths
- Overall, I found the results presented in this paper quite interesting. I am not an expert in complexity analysis so I can fully verify the details of the proof.
- The paper has covered a wide range of models such as smooth GAM,  EBM, NAM, and SHAP.
- The paper also covers a wide range of explanations from Shapley attributes to counterfactual explanations.

Weaknesses
- It might be more convincing to show results with experiments.

---

> ### Author Rebuttal · Authors · 2025-07-30
>
> We thank the reviewer for their valuable and constructive feedback and for acknowledging the significance of our work. See our response below.
>
> **The choice of explanation definitions and why LIME is not analyzed**
>
> The literature offers a wide variety of explanation notions. To reflect this natural diversity, our work considers a broad range of commonly used explanation definitions. These include identifying minimal sufficient feature subsets, generating contrastive explanations that alter the model’s prediction, detecting feature redundancy, and assigning importance scores to features using the widely adopted Shapley value framework. While other explanation notions may also be of interest, we believe that our selection captures a wide array of central facets of how models are typically interpreted. Moreover, a further advantage worth noting of the definitions we chose to study is that they are grounded in formal, well-defined semantics, which enables a rigorous analysis of the computational complexity involved in computing them. These choices are consistent with prior work on the computational complexity of generating explanations for models beyond GAMs (e.g., see [1–5]).
>
> In particular, while LIME is also a popular additive feature attribution method (like SHAP), analyzing its complexity is trickier. This is because LIME constructs a local linear surrogate model using sampled perturbations, a process that lacks a precise and direct formalization suitable for complexity-theoretic analysis. In contrast, SHAP's clear mathematical formulation makes it more suitable for complexity analysis. That said, we agree with the reviewer that studying the complexity of generating local surrogate-based explanations, whether for GAMs or other models, is an interesting and valuable direction. We will mention this as a possible avenue for future research and thank the reviewer for raising this point.
>
> **Follow-up experimental works of our findings**
>
> We thank the reviewer for acknowledging the importance of our theoretical contributions. While our work is indeed theoretical, we believe it is closely tied to practical considerations and can serve as a strong foundation for a wide range of future experimental studies involving GAMs. In particular, we present a broad set of efficient algorithms for computing several highly desirable explanation types and clearly delineate the conditions under which such explanations are tractable or provably hard.
>
> Our analysis also highlights central structural parameters of GAMs that impact computational feasibility — including the input domain (a central theme of our work), model coefficient precision (where quantization can make otherwise intractable problems tractable), and the type of component models used (e.g., splines, neural networks, tree ensembles). We fully agree that exploring experimental applications of these insights is an important direction for future work.
>
> However, as noted by the reviewer as well as other reviewers, our paper already presents a very broad and detailed theoretical analysis across a wide range of complexity-theoretic settings. Due to space constraints, we were not able to include experiments, too, as we believe they could potentially represent a separate line of work on their own. That said, we believe the strong theoretical foundation we provide in this work is comprehensive enough by itself and is well-suited to support such future studies. We will elaborate on some of the future directions we see as valuable in the final version and thank the reviewer for highlighting this point.
>
> [1] Model interpretability through the lens of computational complexity (Barceló et al., Neurips 2020)
>
> [2] On Computing Probabilistic Explanations for Decision Trees (Neurips 2022)
>
> [3] On the Tractability of SHAP Explanations (Van den Broeck et al., JAIR 2022)
>
> [4] On the Complexity of SHAP-Score-Based Explanations: Tractability via Knowledge Compilation and Non-Approximability Results (Arenas et al., JMLR 2023)
>
> [5] Local vs. Global Interpretability: A Computational Complexity Perspective (Bassan et al., ICML 2024)

---

### Decision · Program_Chairs · 2025-09-17

**Decision:**

Accept (poster)

**Comment:**

This paper explores the complexity requirements associated with generating 'explanations' in generalized additive models (GAMs), a popular, expressive class of 'interpretable' models.

The topic is clearly of relevance to the NeurIPS community, and the paper presents novel results.

At the same time, some reviewers express concerns about exposition, and a failure to convince readers of the practical utility of this analysis - or actionable insights arising from it.

Nevertheless, all reviewers return some form of 'accept' in their reviews.  In my post-rebuttal correspondence with them, I asked if any was willing to argue against 'accept'.  None has.

As reviewers largely indicated lower confidence, and did not seem strongly enthusiastic, I have recommended accepting the paper as a poster.